# Covariance resampling for particle filter – state and parameter estimation for soil hydrology

Daniel Berg[1,2], Hannes H. Bauser[1,2], and Kurt Roth[1,3]

[1]Institute of Environmental Physics (IUP), Heidelberg University
[2]HGS MathComp, Heidelberg University
[3]Interdisciplinary Center for Scientific Computing (IWR), Heidelberg University

*Correspondence to:* Daniel Berg (daniel.berg@iup.uni-heidelberg.de)

**Abstract.** Particle filters are becoming increasingly popular for state and parameter estimation in hydrology. One of their crucial parts is the resampling after the assimilation step. We introduce a resampling method that uses the full weighted covariance information calculated from the ensemble to generate new particles and effectively avoids filter degeneracy. The ensemble covariance contains information between observed and unobserved dimensions and is used to fill the gaps between them. The covariance resampling approximately conserves the first two statistical moments and partly maintains the structure of the estimated distribution in the retained ensemble. The effectiveness of this method is demonstrated with a synthetic case – an unsaturated soil consisting of two homogeneous layers – by assimilating time domain reflectometry (TDR)-like measurements. Using this approach we can estimate state and parameters for a rough initial guess with 100 particles. The estimated states and parameters are tested with a forecast after the assimilation, which is found to be in good agreement with the synthetic truth.

## 1 Introduction

Mathematical models represent hydrological and other geophysical systems. They aim to describe the dynamics and the future development of system states. These models need the current state and certain system parameters (e.g. material properties, forcing) for state prediction. Both, state and system parameters, are typically ill-known and have to be estimated.

Data assimilation methods, originally used for state estimation only, are adapted to also estimate parameters and other model components like the boundary condition. The ensemble Kalman filter (EnKF) (Evensen, 1994; Burgers et al., 1998) is a popular data assimilation method in hydrology. It has the advantage of using the ensemble covariance to correlate dimensions with observations to unobserved dimensions. The EnKF with parameter estimation is applied to several hydrological systems. Moradkhani et al. (2005b) used the EnKF for a rainfall-runoff model and Chen and Zhang (2006) for saturated flow modeling. Using a hydrological model based on the Richards equation, the EnKF is mostly applied in synthetic studies (e.g. Wu and Margulis, 2011; Song et al., 2014; Erdal et al., 2015; Shi et al., 2015; Man et al., 2016). However, some applications to real data exist (e.g. Li and Ren, 2011; Bauser et al., 2016; Botto et al., 2018).

As the EnKF is based on Bayes' theorem, all uncertainties have to be represented correctly, otherwise the method has a poorer performance (Liu et al., 2012; Zhang et al., 2015). Nonlinear systems (e.g. systems described by Richards equation) violate the EnKF assumption of Gaussian probability density functions (Harlim and Majda, 2010; DeChant and Moradkhani, 2012). The

dynamics of Richards equation is generally dissipative and the Gaussian assumption is appropriate. However, jumps at layer boundaries, soliton-like fronts during strong infiltration and diverging potentials for strong evaporation deform the probability density function and lead to non-Gaussianity. In this case the probability density function requires higher statistical moments to be described correctly. Particle filter can accomplish this task.

The particle filter has already been used for state and parameter estimation for various hydrological systems. Since parameters do not have their own model dynamics like the state, the resampling step is crucial. Moradkhani et al. (2005a) suggested to perturb the parameters using Gaussian noise with zero mean after the resampling step. They used an unweighted variance of the ensemble modified with a damping factor such that the ensemble spread does not become too large. This method or similar has been used for instance for land surface models (Qin et al., 2009; Plaza et al., 2012), rainfall-runoff models (Weerts and El Serafy, 2006) and soil hydrology (Montzka et al., 2011; Manoli et al., 2015). A common challenge is that with only a rough initial guess, perturbing only the parameters does not guarantee a sufficient ensemble spread and the filter can diverge.

Further development of the resampling for parameter estimation was done by Moradkhani et al. (2012) and Vrugt et al. (2013). They used a Markov chain Monte Carlo (MCMC) method to generate new particles. This method was further used by e.g. Yan et al. (2015) and Zhang et al. (2017). The latter compared the performance of this method with an EnKF and the particle filter presented by Moradkhani et al. (2005a) and found that the performance of the filters were similar with slight advantages for the EnKF. While the MCMC is accurate, it is also expensive, as it requires additional model runs. To increase the efficiency, Abbaszadeh et al. (2018) additionally combined it with a genetic algorithm.

In this paper we introduce the covariance resampling, a resampling method that generates new particles using the ensemble covariance. This method conserves the first two statistical moments in the limit of large numbers while partly maintaining the structure of the estimated distribution in the retained ensemble. With the covariance, the unobserved parameters of the new particles are correlated to the observed state dimensions. The particle filter with covariance resampling is able to estimate state and parameters in case of a difficult initial condition without additional model evaluations, which are necessary for MCMC methods.

## 2 Particle filter

The particle filter is an ensemble-based sequential data assimilation method that consists of a forecast and an analysis step. The ensemble members are called particles. It is used to combine information from observation and model to a posterior estimate. For a detailed review consider e.g. van Leeuwen (2009).

If new information in the form of observations becomes available, the system is propagated forward to the time the observation is taken (forecast). This results in a prior probability density function. The prior is updated with the information of the observation to get the posterior. This is accomplished using Bayes' theorem,

$$P(\boldsymbol{u}|\boldsymbol{d}) = \frac{P(\boldsymbol{d}|\boldsymbol{u})P(\boldsymbol{u})}{P(\boldsymbol{d})} \,, \tag{1}$$

which describes the probability of an event $\boldsymbol{u}$ under the condition of another event $\boldsymbol{d}$. In data assimilation this is used to combine the information of the prior $P(\boldsymbol{u})$ of the state $\boldsymbol{u}$ with the observation $\boldsymbol{d}$. The probability $P(\boldsymbol{d})$ is a normalization constant

$$P(\boldsymbol{d}) = \int d\boldsymbol{u} \, P(\boldsymbol{d}|\boldsymbol{u})P(\boldsymbol{u}) \,. \tag{2}$$

This describes the assimilation process for one observation. For a set of observations $\boldsymbol{d}^{1:k} = (\boldsymbol{d}^1, \boldsymbol{d}^2, \ldots, \boldsymbol{d}^{k-1}, \boldsymbol{d}^k)$, where the superscript denotes a discrete time index, the observations are assimilated sequentially using the recursive filter equation

$$P(\boldsymbol{u}^{0:k}|\boldsymbol{d}^{1:k}) = \frac{P(\boldsymbol{d}^{1:k}|\boldsymbol{u}^k)P(\boldsymbol{u}^k|\boldsymbol{d}^{1:k-1})}{P(\boldsymbol{d}^k)} \,, \tag{3}$$

which follows from Bayes' theorem. The prior distribution at time $k$

$$P(\boldsymbol{u}^k|\boldsymbol{d}^{1:k-1}) = \int d\boldsymbol{u}^{k-1} P(\boldsymbol{u}^k|\boldsymbol{u}^{k-1})P(\boldsymbol{u}^{k-1}|\boldsymbol{d}^{1:k-1}) \tag{4}$$

is calculated by propagating the posterior of the previous analysis $P(\boldsymbol{u}^{k-1}|\boldsymbol{d}^{1:k-1})$ to time $k$ using the transition density $P(\boldsymbol{u}^k|\boldsymbol{u}^{k-1})$.

The particle filter is a Monte Carlo approach, which directly approximates the probability density functions by a weighted ensemble of realizations (particles). This direct sampling allows the particle filter to have non-Gaussian probability density functions. This is in contrast to e.g. the EnKF, which is also based on Bayes' theorem and Monte Carlo sampling but assumes Gaussian distributions.

The posterior distribution of the previous analysis $P(\boldsymbol{u}^{k-1}|\boldsymbol{d}^{1:k-1})$ is approximated by an weighted ensemble of $N$ particles, represented by Dirac delta functions

$$P(\boldsymbol{u}^{k-1}|\boldsymbol{d}^{1:k-1}) = \sum_{i=1}^{N} w_i^k \delta_D(\boldsymbol{u}^{k-1} - \boldsymbol{u}_i^{k-1}) \,. \tag{5}$$

To obtain the new prior $P(\boldsymbol{u}^k|\boldsymbol{d}^{1:k-1})$ for the analysis step at time $k$, it is necessary to solve the integral in Eq. (3). This is achieved by propagating the ensemble forward in time to the next observation using the model equation (forecast). For this, consider the following generic model equation:

$$\boldsymbol{u}^k = \boldsymbol{f}(\boldsymbol{u}^{k-1}) + \boldsymbol{\beta}^k \,, \tag{6}$$

where $\boldsymbol{f}(\cdot)$ is the deterministic part of the model and $\boldsymbol{\beta}^k$ is a stochastic model error.

Using Eq. (3), the weights are updated according to

$$w_i^k = w_i^{k-1} \frac{P(\boldsymbol{d}^k|\boldsymbol{u}_i^k)}{P(\boldsymbol{d}^k)} \,. \tag{7}$$

After the analysis the weights are normalized using the fact that the sum has to be equal to one.

$$\sum_{i=0}^{N} w_i^k \overset{!}{=} 1 \quad \Rightarrow \quad P(\boldsymbol{d}^k) = \sum_{i=0}^{N} w_i^{k-1} P(\boldsymbol{d}^k|\boldsymbol{u}_i^k) \,. \tag{8}$$

In general, $P(\boldsymbol{d^k}|\boldsymbol{u}_i^k)$ is an arbitrary distribution that represents the observation error. We assume Gaussian distributed observation errors which results in:

$$P(\boldsymbol{d}^k|\boldsymbol{u}_i^k) \propto \exp\left[(\boldsymbol{d}^k - \mathbf{H}(\boldsymbol{u}_i^k))^\intercal \, \mathbf{R}^{-1} \, (\boldsymbol{d}^k - \mathbf{H}(\boldsymbol{u}_i^k))\right] \, , \tag{9}$$

where $\mathbf{R}^{-1}$ is the inverse of the observation error covariance and $\mathbf{H}$ is the observation operator that projects the state $\boldsymbol{u}$ from
state-space to observation-space.

To estimate state and parameters simultaneously we use an augmented state. In our case the augmented state $\boldsymbol{u}$ consists of the state $\boldsymbol{\theta}$ (water content) and a set of parameters $\boldsymbol{p}$

$$\boldsymbol{u} = \begin{bmatrix} \boldsymbol{\theta} \\ \boldsymbol{p} \end{bmatrix} . \tag{10}$$

## 3    Resampling

Particle filters tend to filter degeneracy, which is also referred to as filter impoverishment. After several analysis steps, one particle gets all statistical information as its weight becomes increasingly large, whereas the remaining particles only get a small weight such that the ensemble effectively collapses to this one particle. In this case, the filter does not react on new observations and follows the particle with the large weight.

Gordon et al. (1993) introduced resampling to particle filters, a technique that reduces the variance in the weights and has the
potential to prevent filter degeneracy. The idea of resampling is that after the analysis, particles with large weights are replicated and particles with small weights are dropped. This helps that the regions with high weighted particles are represented better by the ensemble, which alleviates the degeneracy of the filter. Filters using resampling are referred to as sequential importance resampling (SIR). There are many different resampling algorithms (see van Leeuwen (2009) for a summary). One of these methods is the stochastic universal resampling.

### 3.1    Stochastic universal resampling

The stochastic universal resampling (Kitagawa, 1996) can be summarized as follows (see also Fig. 1): All weights are aligned after each other on an interval $[0,1]$. A random number in the interval $[0, N^{-1}]$ is drawn from a uniform distribution. This number points to the first particle of the new ensemble, selected by the corresponding weight. Then $N^{-1}$ is added $(N-1)$-times to $x$. Each of the endpoints selects the corresponding particle for the new ensemble. This way some particles get duplicated
and some particles are dropped. With this approach, particles with a weight smaller than $N^{-1}$ can be chosen maximally once, whereas a weight larger than $N^{-1}$ guarantees that the particle is at least chosen once. If all particles have equal weights, no particle is dropped. The result is a new set of $N$ particles. After the resampling step, all weights are set to $N^{-1}$. The stochastic universal resampling has a low sampling noise compared to other resampling methods (van Leeuwen, 2009).

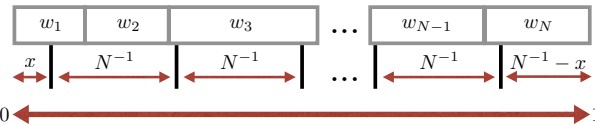

**Figure 1.** Illustration of the universal resampling process. A random number $x$ is drawn from a uniform distribution in the interval $[0, N^{-1}]$. The endpoint of this number indicates the first particle. Then $N^{-1}$ is added $(N-1)$-times to this random number, where every endpoint is a particle of the new ensemble. In the illustration, particle one is chosen once, particle two not once and particle three twice. This way some particles are replicated and other particles are dropped. If the model does not have a stochastic model error, it is necessary to perturb the new particles, otherwise they would be identical and the filter would degenerate.

## 3.2 Covariance resampling

If the model does not have a stochastic model error, like we consider in this study, it is necessary to perturb the particles, otherwise they would be identical and the filter would still degenerate. Even in the presence of a model error it can be useful to perturb the particles after the resampling step. For example if the model error is ill-known or structurally incorrect, it can help to guarantee a sufficient ensemble spread and diversity.

There are different suggestions how to do the perturbation. For example, Moradkhani et al. (2005a) used the ensemble variance to perturb the parameters with a Gaussian with zero mean. Pham (2001) proposed to sample new particles by perturbing the identical particles using a Gaussian with the (damped) ensemble covariance matrix as covariance. Xiong et al. (2006) sampled the whole ensemble from a Gaussian using the first two moments specified by the ensemble (full covariance information), which neglects the particle filter ability to use non-Gaussian distribution. All of these methods have in common, that they alter the estimated distribution to ensure a diverse ensemble.

We neither perturb the duplicated states nor draw a complete new ensemble. The covariance resampling we propose uses the universal resampling – other resampling methods can be equally used – to choose the ensemble members that are kept. Instead of duplicating the particles and setting the weights to $N^{-1}$, the weight of the particles is changed to

$$w_i = \frac{z}{N} \quad \text{with} \quad i \in \{1, 2, .., N'\}, \tag{11}$$

where the particle $i$ is chosen $z$-times and $N'$ is the number of kept particles. In the statistical limit this conserves the estimated distribution.

The total ensemble reduces to $N'$. To have $N$ ensemble members again, $N - N'$ new particles have to be generated. These particles are sampled from a Gaussian $\mathcal{N}\left(\overline{\boldsymbol{u}}, \mathbf{P}^{\mathrm{f}}\right)$ with the weighted mean

$$\overline{\boldsymbol{u}} = \sum_{i=1}^{N} w_i \, \boldsymbol{u}_i \,. \tag{12}$$

and the weighted covariance

$$\mathbf{P}^{\mathrm{f}} = \frac{1}{1 - \sum_{i=1}^{N} w_i^2} \sum_{i=1}^{N} w_i \left[ \boldsymbol{u}_i - \overline{\boldsymbol{u}} \right] \left[ \boldsymbol{u}_i - \overline{\boldsymbol{u}} \right]^{\mathsf{T}} , \tag{13}$$

where the factor $\frac{1}{1 - \sum_{i=1}^{N} w_i^2}$ is Bessel's correction for an unbiased estimate of the weighted covariance. Mean and covariance are calculated using the weights before resampling (Eq. (7)). A weight of $N^{-1}$ is assigned to each of the new particles, which results in a sum of all weights larger than one. Therefore, it is necessary to normalize the weights again. This results in a superposition of the estimated distribution and a Gaussian.

Since the dropped particles are sampled from a Gaussian, the mean and the covariance are conserved in the limit of large numbers. However, the structure of the non-Gaussian distribution is only partly conserved through the retained ensemble. In more difficult situations, where an increasing fraction of particles is resampled, the posterior is dominated by the approximated multivariate Gaussian. However, the approximation allows the use of the covariance information in the ensemble, which facilitates the generation of meaningful new particles and improves the exploration of the state space. In less difficult situations, when only a few particles are resampled, the distribution remains close to the previously estimated, which includes the full structure of the estimated distribution.

Using the multivariate Gaussian utilizes the information of the covariance but sacrifices the more accurate description of the univariate distribution that could be achieved by a kernel density estimation. However, it requires a much smaller sample size compared to a multivariate kernel density estimation.

The whole resampling process is illustrated in Fig. 2. For the pseudocode of the covariance resampling please refer to Appendix A.

New particles are generated with a Cholesky decomposition of the covariance matrix. The calculation of the covariance from the ensemble can result in small numerical errors that have to be regularized, otherwise the decomposition would fail. For details about the generation of new particles and regularization of the covariance matrix see Appendix B.

Pham (2001) introduced a tuning parameter to modify the covariance matrix and Moradkhani et al. (2005a) for the variance, respectively. They used the tuning factor to reduce the amplitude of the perturbation. For the covariance resampling we also introduce a tuning parameter. If the model dynamics does not support a sufficient spread for the ensemble, the perturbation of the covariance resampling has to be large enough to prevent the ensemble from degeneracy. One example for such a case are parameters. The covariance matrix is modified by a multiplicative factor $\gamma$

$$\mathbf{P}'^{\mathrm{f}} = (\boldsymbol{\gamma} \boldsymbol{\gamma}^{\mathsf{T}}) \circ \mathbf{P}^{\mathrm{f}} , \tag{14}$$

where $\circ$ is the entrywise product (Hadamard product). In the case of parameters the factor is chosen larger than one for the parameter space to provide a sufficient ensemble spread.

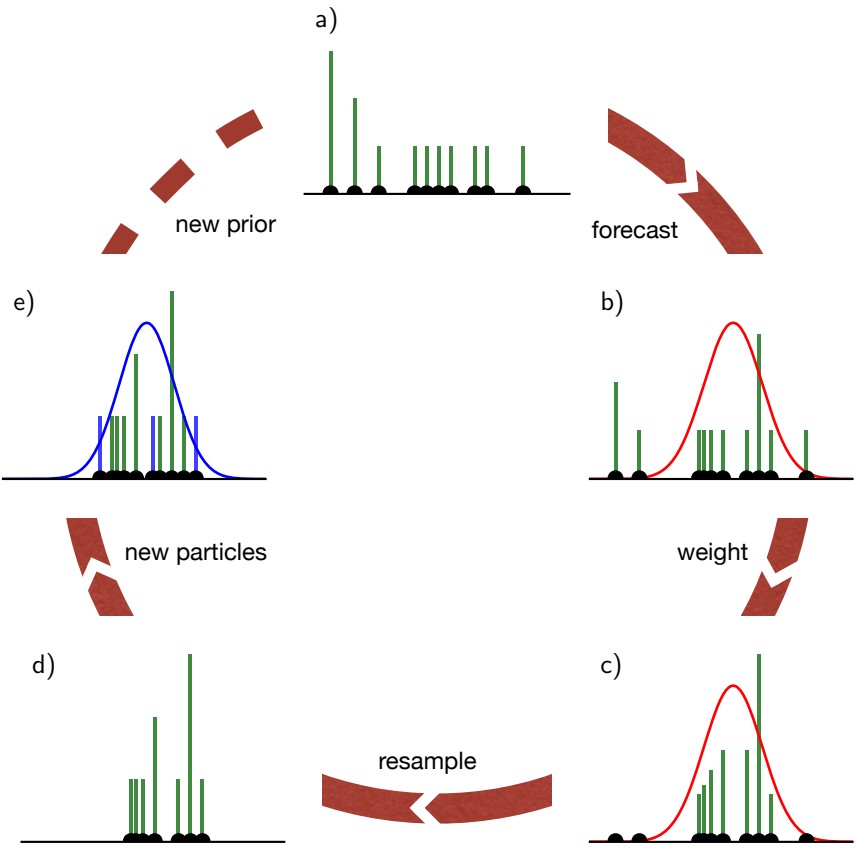

**Figure 2.** Illustration of the particle filter with covariance resampling. The green bars show the weight of each ensemble member (ten in this example) and the black dots the position of it. (a) The prior represented through the ensemble. (b): The ensemble is propagated to the next observation (depicted as Gaussian, red curve). (c): The particles are weighted according to the observation. At this point, some particles have already negligible weight. (d): The universal resampling drops particles with low weight (three in this example). Instead of adding new particles at the same position, only the weights of the kept particles are changed. If a particle is resampled $k$-times, it will get the weight $k\,N^{-1}$. The ensemble size is reduced and new particles have to be added to conserve the ensemble size and avoid filter degeneration. (e): The new particles are drawn from the full covariance of the ensemble (Eq. (13)) and their weight is set to $N^{-1}$. Since new particles with weights are added to the ensemble, it is necessary to normalize the weights to one again. This results in the posterior or the next prior respectively. The pseudocode for the algorithm can be found in Appendix A.

## 4 Case study

The algorithm is tested using a synthetic case study of a one-dimensional unsaturated porous medium with two homogeneous layers. The system has a vertical extent of $1\,\mathrm{m}$ with the layer boundary in the middle at $50\,\mathrm{cm}$. The representation of the considered system is described following the structure of Bauser et al. (2016). The general representation of a system has four

components: *dynamics*, *forcing*, *subscale physics* and *state*. The dynamics propagates the state forward in time, conditioned on the subscale physics and forcing.

The dynamics in an unsaturated porous medium can be described by the Richards' equation

$$\partial_t \theta - \nabla \cdot [K(\theta)[\nabla h_m - 1]] = 0 , \tag{15}$$

where $h_m (\mathrm{L})$ is the matric head, $K(\mathrm{LT}^{-1})$ the isotropic hydraulic conductivity and $\theta (-)$ the volumetric water content. We use the finite-element solver MuPhi (Ippisch et al., 2006) to solve Richards' equation numerically. The resolution is set to $1\,\mathrm{cm}$ which results in a 100-dimensional water content state.

The macroscopic material properties represent the averaged subscale physics with the functions $K(\theta)$ and $h_m(\theta)$ and a set of parameters. In this study, the Mualem-van Genuchten parametrization is used (Mualem (1976), Van Genuchten (1980)):

$$K(\Theta) = K_w\, \Theta^\tau \left[ 1 - \left[ 1 - \Theta^{n/[n-1]} \right]^{1-1/n} \right]^2 , \tag{16}$$

$$h_m(\Theta) = \frac{1}{\alpha} \left[ \Theta^{-n/[n-1]} - 1 \right]^{1-1/n} , \tag{17}$$

with the saturation $\Theta (-)$

$$\Theta := \frac{\theta - \theta_r}{\theta_s - \theta_r} . \tag{18}$$

With these equations the subscale physics is described by six parameters for each layer. The parameter $\theta_s (-)$ is the saturated water content and $\theta_r (-)$ the residual water content. The matric head $h_m$ is scaled with the parameter $\alpha (\mathrm{L}^{-1})$ that can be related to the air entry value. The parameter $K_w (\mathrm{LT}^{-1})$ is the saturated hydraulic conductivity, $\tau (-)$ a tortuosity factor and $n (-)$ is a shape parameter. In this study the parameters $\alpha, n$ and $K_w$ will be estimated for each layer. Combining Eq. (17) and Eq. (16) results in a conductivity function

$$K(h_m) = K_w \left[1 + (\alpha h_m)^n\right]^{-\tau(1-1/n)} \left[1 - (\alpha h_m)^{n-1}(1 + (\alpha h_m)^n)^{-1+1/n}\right]^2 \tag{19}$$

that incorporates all estimated parameters.

For the true trajectories and the observations, parameters by Carsel and Parrish (1988) for loamy sand are used for the upper layer (layer 1) and sandy loam for the lower layer (layer 2). Table 1 gives the true values for the estimated parameters and Table 2 the values for the fixed parameters, respectively. In the following the parameters will have an index representing their corresponding layer.

Since state and parameters are estimated, we separate the model equation Eq. (6) into

$$\mathbf{u}^n = \begin{bmatrix} \boldsymbol{\theta}^k \\ \boldsymbol{p}^k \end{bmatrix} = \begin{bmatrix} \boldsymbol{f}(\boldsymbol{\theta}^{k-1}, \boldsymbol{p}^{k-1}) \\ \boldsymbol{p}^{k-1} \end{bmatrix} , \tag{20}$$

with a constant model for the parameters $\boldsymbol{p}$ and Richards' equation as $f(\cdot)$. Note that the model error of equation Eq. (6) is set to zero. In hydrology the model error is typically ill-known and can vary both in space and time.

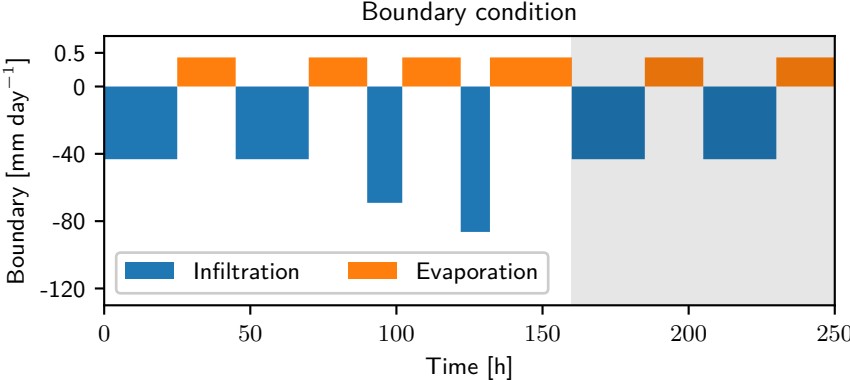

**Figure 3.** Upper boundary condition for the data assimilation case. Four rain events (blue) followed by a dry period (orange) are used for the test of the data assimilation run. After this run, two additional rain events and dry periods are used in a free run to test the assimilation results (grey background). The intensity and duration of these events is set equal to the first events of the data assimilation run. Note the different axes for infiltration and evaporation.

The forcing is reflected in the boundary condition of the simulation. For the lower boundary, a Dirichlet condition with zero potential (groundwater table) is used. The upper boundary condition is chosen as a flux boundary (Neumann), representing four rain events with increasing intensity and dry periods in between (see Fig. 3).

Using infiltrations with an increasing intensity has the advantage that the system has more time to adjust the parameters. This way less observations are necessary to resolve the infiltration front and the information of the observations can be incorporated in the state and parameters. The stronger infiltration front in the end is used to test the robustness of the estimated state and parameters.

The last component of the system is the state. Initially, the system is in equilibrium and will be forced by the boundary condition. The initial state is depicted in Fig. 4. Six TDR-like observations are taken equidistantly in each layer at the positions $(0.1, 0.25, 0.3)$ m for layer 1 and $(0.6, 0.75, 0.9)$ m for layer 2. The measurement error is chosen to be $\sigma_{\text{Obs}} = 0.007$ (e.g. Jaumann and Roth, 2017). Observations are taken hourly for the duration of $160$ h.

For the initial state of the data assimilation, the observations at time zero are used. The measured water content is interpolated linearly between the measurements and kept constant towards the boundary. Additionally, the saturated water content for loamy sand, which is $0.41$ is taken as the lower boundary. The approximated state is used as the ensemble mean for the particle filter. This procedure is a viable option for real data although it represents a rather crude approximation of the real initial condition.

The approximated state is perturbed by a correlated multivariate Gaussian. The main diagonal of the covariance matrix is $0.003^2$. The variance is chosen such that the ensemble represents the uncertainty of the water content in most parts (see Fig. 4). The off-diagonal entries are determined by the following two steps: (i) All covariances between the two layers are set to zero to ensure no correlations across the layer boundary. (ii) The remaining entries are the variance of the main diagonal multiplied

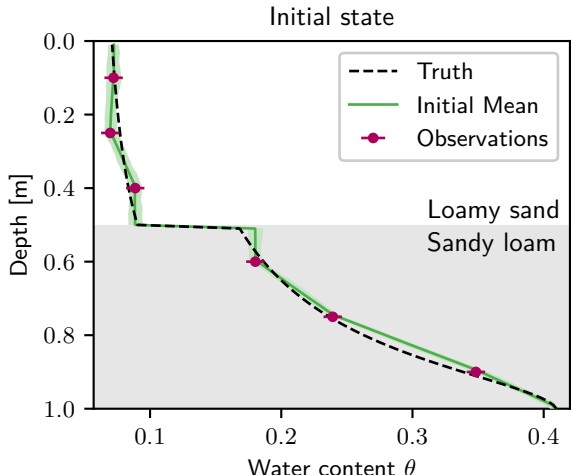

**Figure 4.** Initial state for the data assimilation run. Observations (purple) at time zero are connected linearly and set constant towards the layer and upper boundary. For the lower boundary, the saturated water content $\theta_r = 0.41$ of sandy loam is used for the interpolation. The ensemble with 100 ensemble members is generated by perturbing the interpolated state using a spatially correlated Gaussian. The 95 %-quantile of the initial ensemble is shown in light green. The initial truth that is used for the observations (purple) is shown as a black dashed line.

**Table 1.** True Mualem-van Genuchten parameters and range of the uniformly distributed initial guess.

| Parameter | Truth | Lower | Upper |
|---|---|---|---|
| $n_1$ $[-]$ | 2.28 | 2.2 | 3.5 |
| $n_2$ $[-]$ | 1.89 | 1.8 | 3.2 |
| $\alpha_1$ $[\mathrm{m}^{-1}]$ | -12.4 | -14 | -12 |
| $\alpha_2$ $[\mathrm{m}^{-1}]$ | -7.5 | -10.5 | -6.5 |
| $\log_{10}(K_{w,1})$, $K_w$ in $[\mathrm{m\,s}^{-1}]$ | -4.40 | -7 | -4 |
| $\log_{10}(K_{w,2})$, $K_w$ in $[\mathrm{m\,s}^{-1}]$ | -4.91 | -7.5 | -4 |

with the Gaspari and Cohn function (Gaspari and Cohn, 1999). The distance for the Gaspari and Cohn function is the distance of the off-diagonal entry from the main diagonal and a length scale of $c = 10\,\mathrm{cm}$ is used. This way, the water content is only correlated in the range of $20\,\mathrm{cm}$.

The use of the covariance increases the diversity of the ensemble and also helps to avoid degeneration. Using uncorrelated Gaussian random numbers with zero mean would result in a fast degeneration of the particle filter caused by the dissipative nature of the system. The perturbation would simply dissipate and the ensemble collapses.

The initial parameters for the ensemble are uniformly distributed. The ranges of the uniform distributions are given in Tab. 1. Note that the decadic logarithm of the saturated conductivity $K_w$ is used for the estimation, so $K_w$ spans three orders of

**Table 2.** Fixed Mualem-van Genuchten parameters.

| Parameter | Layer 1 | Layer 2 |
|---|---|---|
| $\theta_s\ [-]$ | 0.41 | 0.41 |
| $\theta_r\ [-]$ | 0.057 | 0.065 |
| $\tau\ [-]$ | 0.5 | 0.5 |

magnitude. The filter can also estimate the state and parameters for an extended range. In this study, the ensemble size is $100$. Increasing the initial uncertainty of the parameters, increases the complexity of the problem and the filter needs more ensemble members to converge. The multiplicative factor Eq. (14) is set to:

$$\gamma = \begin{bmatrix} \gamma_{\theta,100} \\ \gamma_{p,6} \end{bmatrix}, \tag{21}$$

where $\gamma$ is separated to $\gamma_\theta$ and $\gamma_p$ for the water content and the parameter, respectively. The number in the subscript denotes the dimension of the factor. The covariance in the 100-dimensional state space is unchanged. For the parameter space a factor of 1.2 is used to compensate the missing dynamics. The subscript for the dimension will be omitted in the following.

After the assimilation, the ensemble is used to run a forecast without data assimilation and the mean is calculated from the propagated ensemble using the weights of the last assimilation time. In this run two additional infiltration events and evaporation periods (see Fig. 3) are used to test the forecasting ability of the estimated states and parameters.

## 5 Results

The development of the parameters is depicted in Fig. 5. The saturated conductivity $K_{w,1}$ (Fig. 5a) can be estimated quickly because the infiltration front induces dynamics in the first layer which is strongly dependent on $K_w$. Instead, $K_{w,2}$ (Fig. 5b) is not sensitive to the dynamics in the first hours, as the infiltration front did not reach the second layer yet. At around $46\,\mathrm{h}$, the infiltration front reaches the first observation position in the second layer and the estimation for $K_{w,2}$ improves quickly.

If dynamic is induced in the system, the ensemble spread in the water content space increases because of different parameter sets. This makes the particles and their corresponding parameter sets distinguishable and parameter estimation possible. The parameters $n_1$ and $n_2$ (Fig. 5c and d) as well as $\alpha_2$ (Fig. 5f) can be estimated well. One exception is $\alpha_1$ (Fig. 5e). This parameter is insensitive to the observations. The effect of $\alpha$ on the trajectory of the ensemble members is limited to a small region next to the layer boundary. Further away, wrong values can be compensated by $n$. The parameter $\alpha_1$ jitters randomly around a value slightly larger than the truth. In a synthetic case, the estimation of $\alpha_1$ can be improved easily by having observations directly next to the boundary. This way a different value for $\alpha_1$ would have a direct influence on the trajectory and $\alpha_1$ would become sensitive to the observations. However, in reality it is not feasible to change the measurement position or measure directly next to the layer interface. We decided to retain these positions to illuminate an often encountered practical difficulty.

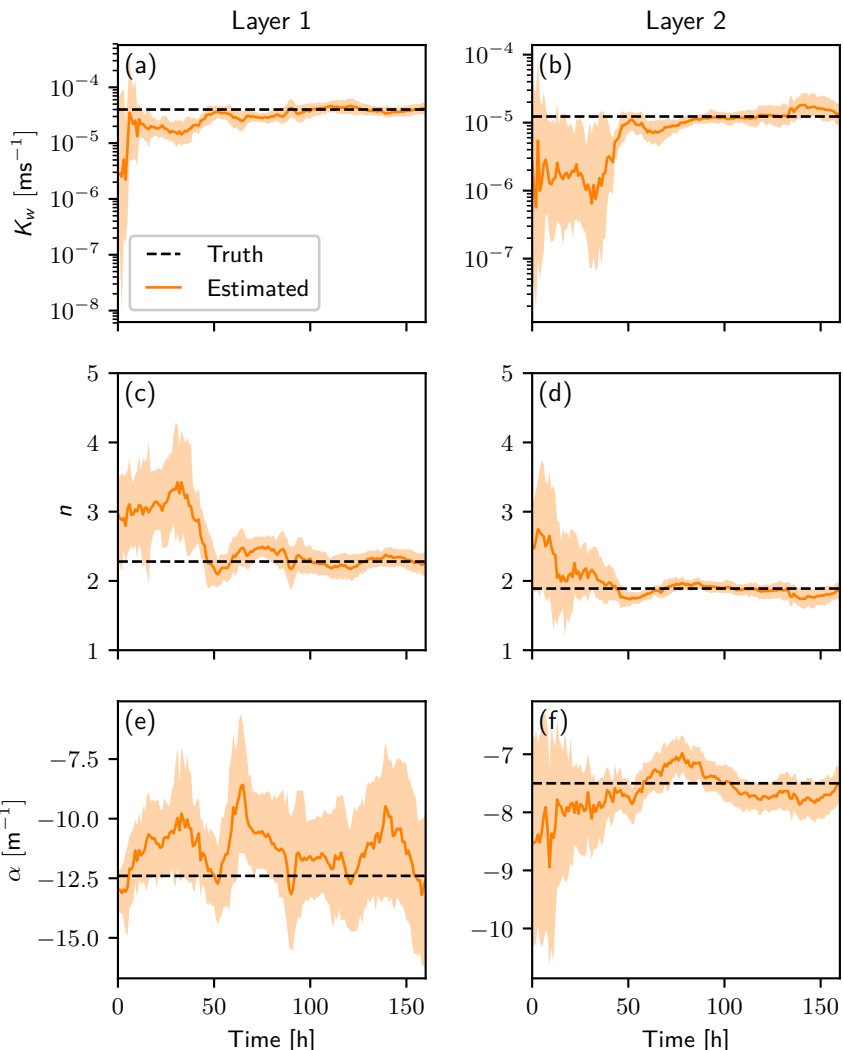

**Figure 5.** Estimation of all six parameters ((a): $K_{w,1}$, (b): $K_{w,2}$, (c): $n_1$, (d): $n_2$, (e): $\alpha_1$, (f): $\alpha_2$) over time. The ensemble mean is shown in orange and the 95 %-quantile of the ensemble in light orange. The truth is a dashed black line.

To see the effect of the parameters on the forward propagation, it is necessary to have a closer look at the conductivity function Eq. (19). This function is used for the model forward propagation and many parameter sets can effectively describe the same situation in a limited regime of the hydraulic head. The function is shown in Fig. 6 for both layers for the prior and the final parameters. The difference between the truth and the estimated parameters is small even though the 95 %-quantile of the prior ensemble does not include the truth for small $h_m$ for layer 1.

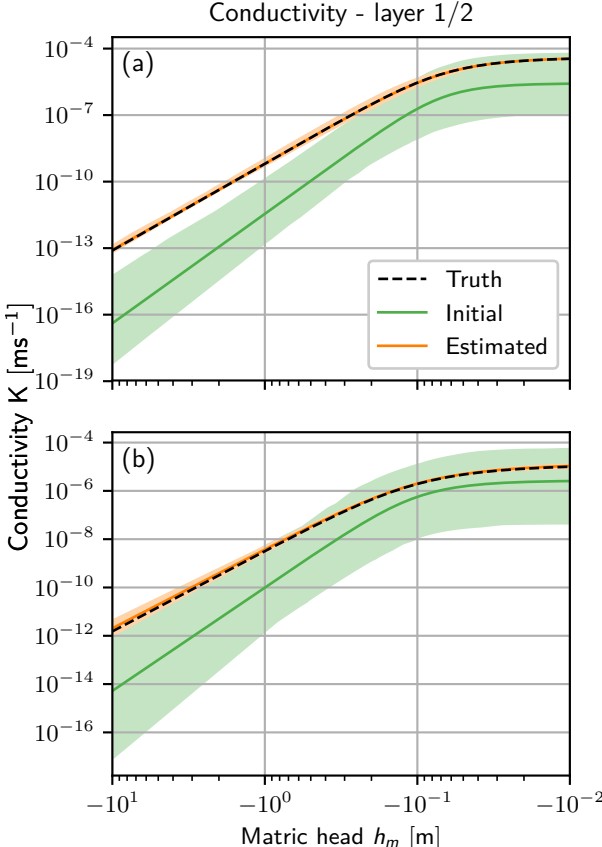

**Figure 6.** Conductivity function $K(h_m)$ (Eq. (19)) for (a): layer 1 and (b): layer 2. In this function all estimated parameters are represented. The initial 95 %-quantile of the ensemble (light green) with the mean (green) are shown. The truth (dashed black) is almost congruent with the estimated mean (orange), such that only the 95 %-quantile of the final ensemble (light orange) is visible.

The final water content state estimated with the particle filter agrees with the synthetic truth in a narrow band, while the mean of the ensemble propagated without data assimilation is far off (see Fig. 7). The estimated state is slightly biased to higher water contents. We checked that the direction of the bias is a random effect and is different for different seeds. The observation of a bias instead is caused by long range correlations of the model. In this case, the system has started to relax after the last infiltration and a higher water content in one part results in a higher water content in the rest of the layer. The ensemble spread next to the layer boundary is caused by the large spread of $\alpha_1$.

To analyze the ensemble, we take a closer look at the effective sample size and the number of particles that are resampled. The effective sample size is defined as (Doucet, 1998):

$$N_{\text{eff}} = \frac{1}{\sum_{i=1}^{N} w_i^2} \tag{22}$$

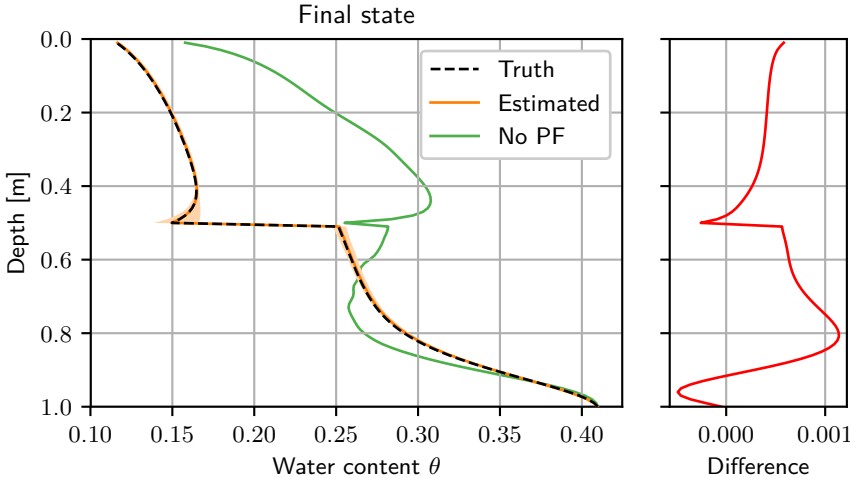

**Figure 7.** Final water content state after the assimilation run. The truth (dashed black) is almost congruent with the estimated mean (orange), such that only the 95 %-quantile of the ensemble (light orange) is visible. The final ensemble with the corresponding weights is used to start a free forward run afterwards. The mean without data assimilation (green) is calculated from a forecast of the initial ensemble (see Fig. 4 and Fig. 6). The difference of the estimated water content and the synthetic truth lies in a narrow band, with a small bias to larger water contents.

and gives an estimate of how many particles are significant. For example, if one particle accumulates all the weight $N_{eff} = 1$ and the ensemble is effectively described only by this particle.

Fig. 8 shows the effective sample size and the number of new particles over time. The effective sample size drops every time new information is available and the number of resampled particles increases. For times $t < 15h$, the effective sample
size drops to small values. The infiltration front propagates through the first layer, which leads to a large ensemble spread caused by unknown parameters and only a few particles have a significant weight. The filter assimilates the information from the observations and resamples particles with low weight. This improves the state and parameters and leads to an increasing effective sample size until the infiltration front reaches the second layer ($t \approx 46h$). The effective sample size decreases rapidly because the parameters in the second layer are still unknown and lead to a large ensemble spread again. This variation of the
effective sample size occurs every time the filter gets new information that leads to a discrepancy between states of the particles and the observations.

The effective sample size is a crucial parameter for the covariance resampling. If it drops to low values the filter can degenerate because effectively too few particles contribute to the weighted covariance (Eq. (13)) and the covariance information becomes insignificant.
For further analysis, the RMSE is calculated based on the difference of the true water content and the weighted mean at every observation time. This includes also the unobserved dimensions. The RMSE shows a similar behavior as the parameters and the effective sample size (see Fig. 9). During the first infiltration, the state and the parameters are improved, the RMSE

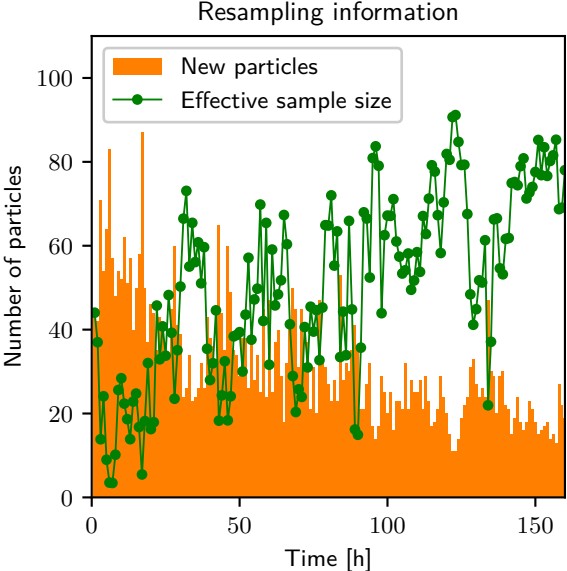

**Figure 8.** Amount of particles that are resampled (orange) and the effective sample size (green dots). The lines connecting the dots are for better visibility of the time dependent behavior. The effective sample size increases while the number of resampled particles decreases. Every infiltration reduces the effective sample size and leads to more resampled particles.

becomes smaller. Then the infiltration front reaches the boundary interface. The parameters of the second layer and $\alpha_1$ are still wrong and diverse. This leads to a spread of the ensemble by the system dynamics and also a shift of the mean away from the truth. The parameters in the second layer are estimated and the state is corrected, which leads to a decrease in the RMSE. The increase for the next infiltration events becomes smaller since state and parameters are already improved.

5    After the data assimilation, the final ensemble including the weights are used for a forecast. This forward run without data assimilation shows that the RMSE oscillates in a narrow range. These oscillations are part of the unobserved space next to the boundary and are mainly caused by the wrong value of $\alpha$ for the first layer. In the beginning, the RMSE stays small, but when the infiltration front reaches the boundary of the two layers, the mean state and the truth begin to deviate from each other (limited to the boundary interface). After the infiltration front passed, the state starts to equilibrate and is increasingly defined

10   by the whole parameter set, which has a certain distance to the true equilibrium.

## 6   Practical considerations

For the presented case study, this section explores two issues relevant when applying the proposed covariance resampling method: (i) the choice of the factor $\gamma$ in interplay with the ensemble size for different seeds and (ii) the effect of a model bias, in our case simulated using a biased upper boundary condition.

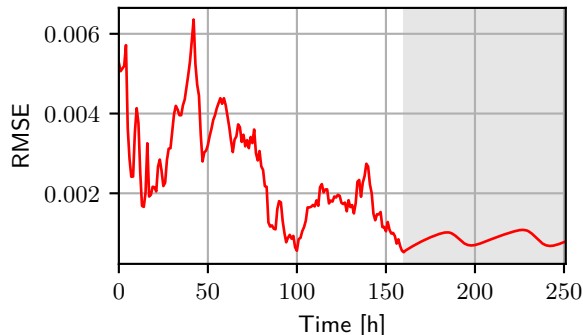

**Figure 9.** The RMSE (red) of the water content calculated between the truth and the estimated mean including all dimensions. After 160 hours the free run starts (grey background). The mean of the free run is calculated using the propagated ensemble members with their corresponding weights at the last assimilation time. During this time, the RMSE is about $10^{-3}$. For the assimilation and the free run the RMSE increases with each infiltration.

## 6.1 Tuning parameter $\gamma$

To explore the convergence of the particle filter with covariance resampling, the filter was run for $40$ different seeds, varying ensemble sizes and for four different tuning parameters $\gamma$ (see Eq. (14)). The tuning parameter is only changed for the parameter space $\gamma_p$, while in state space the same value is used as in the case study ($\gamma_\theta = 1.0$). Four different tuning parameters are used: $\gamma_p = 1.0$ (no change in the covariance), $\gamma_p = 1.1$, $\gamma_p = 1.2$ (also used in the case study) and $\gamma_p = 1.3$. The remaining setup of the system (e.g. initial condition, boundary condition) is identical as in Sect. 4.

Figure 10a shows the RMSE of the water content, calculated between the truth and the estimated mean at the last observation time. The RMSE is averaged over the 40 different seeds. For small ensemble sizes the filter degenerates for every value of $\gamma_p$, which leads to a large RMSE. Except for the case $\gamma_p = 1.0$, the RMSE converges for less then 200 ensemble members to a common value independent of the tuning parameter. For $\gamma_p = 1.0$ the RMSE approaches this value as well, but does not reach it completely even for 1000 ensemble members. While the use of the tuning factor is not mandatory, increasing $\gamma_p$ to a value slightly larger then one reduces the necessary ensemble size by an order of magnitude.

Figure 10b shows the standard deviation $\sigma$ of the ensemble in water content space at the last observation time, averaged over the 40 different realizations. For small ensemble sizes the filter degenerates for most of the 40 runs. In this case the standard deviation is zero. Increasing the ensemble size, increases the number of successful runs and the standard deviation converges to a final value. The convergence is similar to the convergence of the RMSE in Fig. 10a. However, the ensemble converges to different $\sigma$ for different values of $\gamma_p$. The tuning factor affects the covariance of the newly generated particles and thus an increasing factor results in an increased variance in the estimated distribution. The standard deviation of the ensemble is overestimated for $\gamma > 1$. The mean is not influenced for the chosen values of $\gamma_p$. However, increasing the value further will

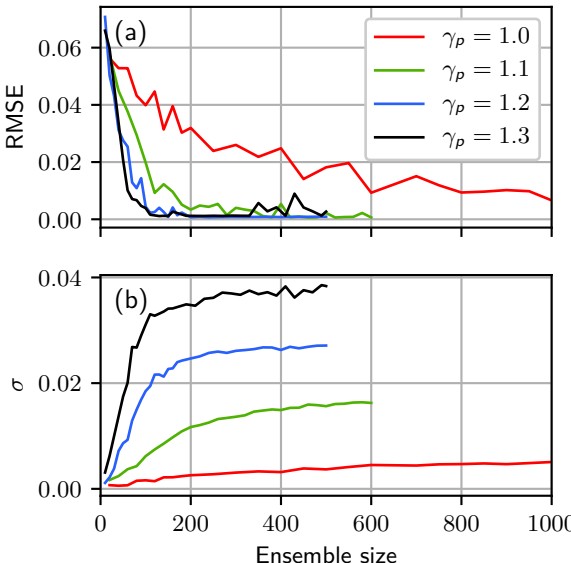

**Figure 10.** The RMSE and the standard deviation of the water content at the last observation time (160 h), averaged over the 40 different realizations. The RMSE is calculated between the truth and the estimated mean. Both quantities are shown for varying factors of $\boldsymbol{\gamma}_p$ (Eq. (14)): $\boldsymbol{\gamma}_p = 1.0$ (red), $\boldsymbol{\gamma}_p = 1.1$ (green), $\boldsymbol{\gamma}_p = 1.2$ (blue) and $\boldsymbol{\gamma}_p = 1.3$ (black). Note the different scaling of the x-axes.

eventually increase the uncertainty too strong and influence the estimation itself (see Supplementary). For an analysis of the estimated mean for the saturated conductivity in the second layer please refer to Appendix C.

The tuning factor has similarities to multiplicative inflation for the EnKF (Anderson and Anderson, 1999). It increases the uncertainty and reduces filter degeneracy. However, the simple choice of a constant multiplicative factor $\gamma$ can lead to too large
uncertainties. For an better uncertainty estimation it is necessary to set $\gamma = 1$. This requires a larger ensemble size. Therefore, an adaptive factor similar to the EnKF (e.g. Wang and Bishop, 2003; Anderson, 2007; Bauser et al., 2018) is desirable to increase the efficiency of the filter further and to achieve a better uncertainty representation of the ensemble.

## 6.2  Model error

Model errors are omnipresent in real systems. They can have a structural or stochastic nature, different intensities, and they
can manifest e.g. by biases in the results. For data assimilation of real measurements, the consideration of model errors is an important part for the success of the methods. Several extensions and modifications to sequential data assimilation methods have been discussed (e.g. Li et al. (2009), Whitaker and Hamill (2012) and Houtekamer and Zhang (2016)) to compensate and improve the filter performance in presence of model errors.

In the course of this paper, we briefly study the behavior of the particle filter with covariance resampling for the case of a
biased upper boundary condition. Two cases are considered, one with a $10\%$ and one with a $20\%$ bias towards less water. This

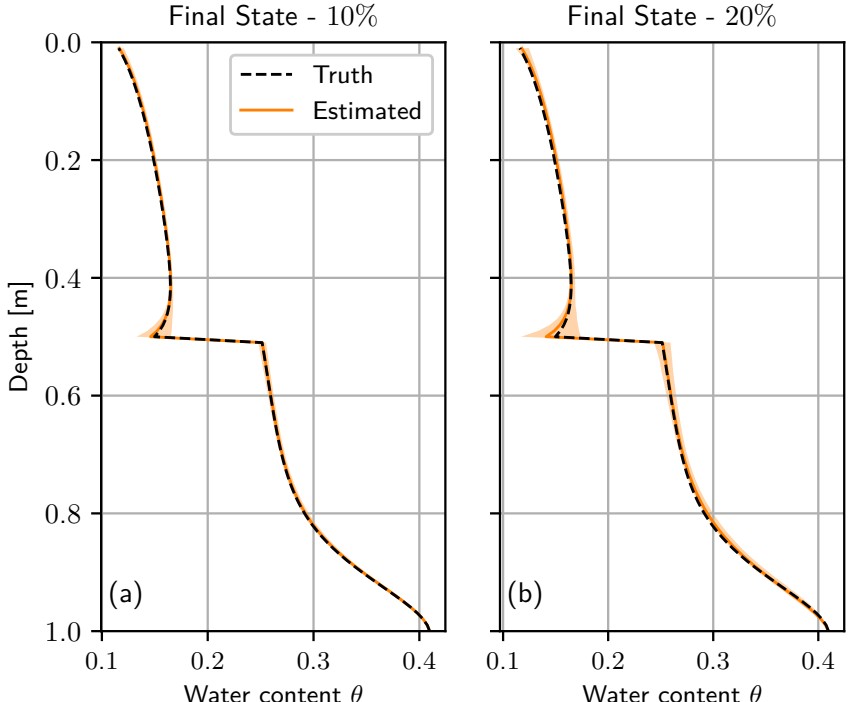

**Figure 11.** Final water content state after assimilation run using a bias for the upper boundary condition of (a): 10% and (b): 20%. The truth (dashed black) is almost congruent with the estimated mean (orange). The light orange areas represent the 95 %-quantile of the ensemble of (a): 600 and (b): 1200 ensemble members.

means the amount of rain is reduced and the evaporation is increased by these percentages. The observations are still generated using the previous boundary condition (Fig. 3). This means that the ensemble is propagated with a biased model, compared to the truth, for the complete assimilation run.

Except of the ensemble size and the upper boundary condition the setup is identical as in Sect. 4. To achieve converging
5    results with $\gamma_\theta = 1.0$ and $\gamma_p = 1.2$, the ensemble size is increased to 600 and to 1200 ensemble members for the case of the 10% and the 20% bias, respectively. By increasing the tuning factor $\gamma$ for the state to $\gamma_\theta = 1.1$ the necessary ensemble size can be reduced to 300 (10%) and 600 (20%). This artificially increases the uncertainty in state space, which helps the filter to compensate the bias during estimation. For better comparison with the presented case study in Sect. 4, we show the results for the case with $\gamma_\theta = 1.0$ and $\gamma_p = 1.2$ in the following.

10    Figure 11 shows the final estimated state and the ensemble. The variance of the ensemble is larger compared to the case that uses the true boundary condition (see Fig. 7). The bias in the boundary condition leads to a larger uncertainty in state estimation, which increases with increasing bias (compare Fig. 11a and Fig. 11b). Although the difference to the mean slightly increases, the estimated mean still matches the truth well.

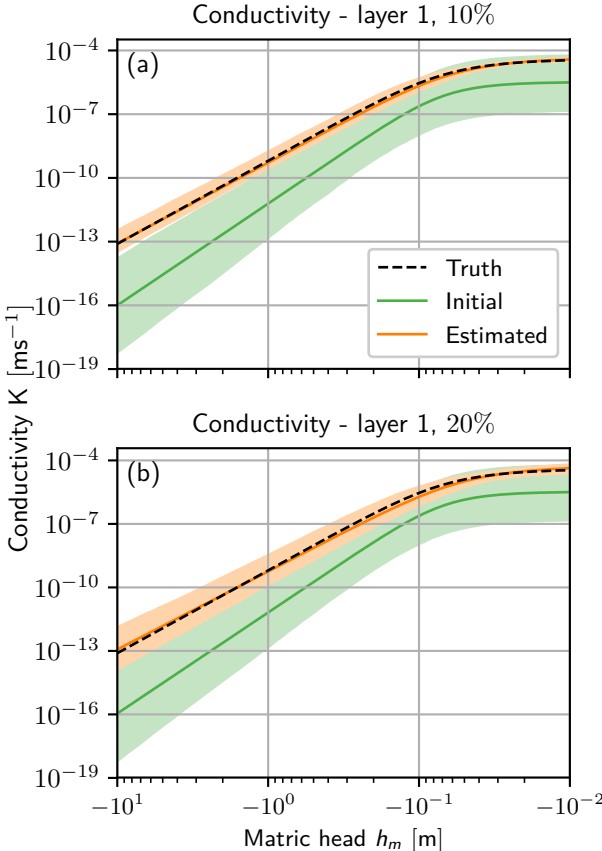

**Figure 12.** Conductivity function $K(h_m)$ (Eq. (19)) for a bias of (a): 10% and (b): 20%. The initial 95%-quantile of the ensemble (light green) with the mean (green) are shown. The truth (dashed black) is almost congruent with the estimated mean (orange), such that only the 95%-quantile of the final ensemble (light orange) is visible. For (a): 600 and (b): 1200 ensemble members are used.

The conductivity function (see Fig. 12) shows a similar behavior as the state. Compared to the case using the true boundary condition (see Fig. 6), the ensemble spread is larger, which increases with the bias in the boundary condition (compare Fig. 12a and Fig. 12b). The biased upper boundary condition leads to a bias in the conductivity function, which is not perfectly visible due to the logarithmic scale. The bias in the conductivity is larger for an larger error in the boundary condition. This behavior is also found for the conductivity function in the second layer.

## 7 Summary and conclusions

We introduced a resampling method for particle filters that uses the covariance information of the ensemble to generate new particles and effectively avoids filter degeneracy. The method was tested in a synthetic one-dimensional unsaturated porous

medium with two homogeneous layers. Even with just a rough initial guess, a broad parameter range and only 100 ensemble members, the estimation shows excellent results. After the assimilation, the results are verified in a free run with the final results.

The covariance connects information between observed and unobserved dimensions. This has some similarity to the ensemble Kalman filter but in our approach information of the non-Gaussian distribution is partly maintained in the retained ensemble. Even tough the RMSE of the water content includes the unobserved state dimensions, it stays in a narrow range (RMSE is about $10^{-3}$) during the forecast. With every infiltration, the RMSE shows excursions caused by a wrong value of parameter $\alpha$ in the first layer that results in a wrong state near the layer boundary during the infiltration.

Transferring the information to the unobserved dimensions helps the filter to not degenerate when only a rough initial guess is available. The states and parameters are both altered actively. For the used initial condition, perturbing the parameters only (Moradkhani et al., 2005a), can lead to filter degeneracy because the state is only changed by the dynamics of the system. Compared to the particle filter with MCMC resampling (Moradkhani et al., 2012; Vrugt et al., 2013), the covariance resampling presented in this study has the advantage that it does not need additional model runs to generate new particles. However, the covariance resampling has to calculate the covariance matrix and perform a Cholesky decomposition every assimilation step. Similar to localization for the ensemble Kalman filter (Houtekamer and Mitchell, 2001; Hamill et al., 2001), it is possible to localize the covariance in the resampling to increase the efficiency.

The effective sample size (Eq. (22)) is a crucial parameter for this method. The covariance resampling needs a sufficient effective sample size, otherwise the calculation of the covariance matrix (Eq. (13)) becomes inaccurate and the filter may degenerate. In such a situation, the filter can be improved by resetting the weights to $N^{-1}$ or increasing the ensemble size. In our example this was not necessary because the effective sample size was critical only for single assimilation steps.

The filter performance can be increased by a tuning parameter $\gamma$. The tuning parameter can significantly reduce the necessary ensemble size but has to be chosen carefully because otherwise the covariance can be overinflated. The mean is independent of the chosen $\gamma$, however, for $\gamma > 1$ the ensemble uncertainty is overestimated. In the presented case study, the tuning parameter reduced the necessary ensemble size by an order of magnitude. For cases with a model error, using the tuning parameter also for the state dimensions can be beneficial to stabilize the filter and reduce the necessary ensemble size further.

Different parameter sets can approximately describe the same conductivity function (Eq. (19)) in a certain matric head regime. Model dynamics is necessary to differentiate between those sets. If the infiltration covers only a small regime, the conductivity function is only significant in the observed range and can differ from the truth otherwise. This is also reflected in the chosen boundary condition. Starting with infiltrations with low intensity but longer duration helps the filter to explore the water content range slowly and the observations can resolve the infiltration front.

The covariance resampling connects observed with unobserved dimensions to effectively estimate parameters and prevent filter degeneracy. It conserves the first two statistical moments in the limit of large numbers, while partly maintaining the structure of the non-Gaussian distribution in the retained ensemble. The method is able to estimate state and parameters in case of a difficult initial condition without additional model evaluations and using a rather small ensemble size.

## Appendix A: Pseudocode

The following pseudocode describes the covariance resampling for a single time $k$, where the propagated ensemble and the calculated weights are given.

---

**Algorithm 1** Pseudocode for the covariance resampling

---

**Require:** weights $w_i^k$ at observation time $t_k$ and the ensemble of $N$ states $\boldsymbol{u}_i^k$

**(a):** compute weighted ensemble covariance $\mathbf{Q}$ (Eq. (13))

**(b):** determine eigenvalues $\{\lambda_1, \lambda_2, \ldots, \lambda_N\}$ of $\mathbf{Q}$

**(c):** if necessary regularize $\mathbf{Q}$:

    **if** $\min(\{\lambda_1, \lambda_2, \ldots, \lambda_N\}) < 0$ **then**

        $\mathbf{Q}_{\text{Reg.}} = \mathbf{Q} + \lambda_{\max}\mathbf{I}$    // $\lambda_{\max} \approx |\min(\{\lambda_1, \lambda_2, \ldots, \lambda_N\})|$

    **end if**

**(d):** optional: multiply $\mathbf{Q}$ with the tuning parameter $\boldsymbol{\gamma}$ (see Eq. (14))

**(e):** universal resampling to determine number of child particles $z$ (different method can be used for selection):

    draw random number $x$ from uniform distribution $U(0, N^{-1})$

    **for** $i = 0$ **to** $i < N$ **do**

        $l = \sum_{m=0}^{i} w_m^k$

        $z_i = 0$

        **while** $x < l$ **do**

            $z_i = z_i + 1$

            $x = x + N^{-1}$

        **end while**

    **end for**

**(f):** generate new particles:

    **for** $i = 0$ **to** $i < N$ **do**

        **if** $z_i > 0$ **then**

            keep particle $i$

            assign $w = \frac{z}{N}$ to this particle

            generate $z_i - 1$ particles using $\boldsymbol{u}_i$ and $\mathbf{Q}$

            assign $w = N^{-1}$ to these new particles

        **end if**

    **end for**

**(g):** renormalize weights $w_i = \frac{w_i}{c}$    // $c = \sum_{i=0}^{N} w_i$

---

## Appendix B: Generation of correlated random numbers

### B1 Cholesky decomposition

Correlated random numbers are generated using the Cholesky decomposition. We use the LDLT decomposition which is part of the *Eigen3* library (Guennebaud, Jacob et al., 2010). Decomposing the covariance matrix $\mathbf{Q}$ leads to

$$\mathbf{Q} = \mathbf{LDL}^\intercal, \tag{B1}$$

where $\mathbf{D}$ is a diagonal matrix and $\mathbf{L}$ is a lower unit triangular matrix. The LDLT form of the decomposition is related to the LLT-form by

$$\mathbf{Q} = \mathbf{L}'\mathbf{L}'^\intercal \quad \text{with} \quad \mathbf{L}' := \mathbf{LD}^{\frac{1}{2}}. \tag{B2}$$

To draw a random vector $\mathbf{y}$ from a Gaussian distribution $\mathcal{N}(\boldsymbol{\mu}, \mathbf{Q})$ with mean $\boldsymbol{\mu}$, we first generate a normal distributed ($\mathcal{N}(\mathbf{0}, \mathbf{I})$) random vector $\mathbf{x}$. This vector is multiplied with $\mathbf{L}'$ and the mean $\boldsymbol{\mu}$ is added:

$$\mathbf{y} = \mathbf{L}'\mathbf{x} + \boldsymbol{\mu} \tag{B3}$$

To verify that this gives the correct result the covariance matrix of $\mathbf{y}$ is calculated:

$$\overline{(\mathbf{y} - \boldsymbol{\mu})(\mathbf{y} - \boldsymbol{\mu})^\intercal} = \overline{\mathbf{L}'\mathbf{x}(\mathbf{L}'\mathbf{x})^\intercal} = \mathbf{L}'\overline{\mathbf{x}\mathbf{x}^\intercal}\mathbf{L}'^\intercal = \mathbf{L}'\mathbf{I}\mathbf{L}'^\intercal = \mathbf{Q} \tag{B4}$$

yields $\mathbf{Q}$ as required.

### B2 Regularization of the ensemble covariance matrix

The calculation of the Cholesky decomposition (LDLT-version) is only possible if the matrix is not indefinite. Mathematically, a covariance matrix has to be positive semidefinite:

$$\boldsymbol{v}^\intercal \mathbf{Q} \boldsymbol{v} = \boldsymbol{v}^\intercal \sum (\boldsymbol{y}_i - \boldsymbol{\mu})(\boldsymbol{y}_i - \boldsymbol{\mu})^\intercal \boldsymbol{v} \tag{B5}$$

$$= \sum \boldsymbol{v}^\intercal (\boldsymbol{y}_i - \boldsymbol{\mu})(\boldsymbol{y}_i - \boldsymbol{\mu})^\intercal \boldsymbol{v} \tag{B6}$$

$$= \sum (\boldsymbol{v}^\intercal (\boldsymbol{y}_i - \boldsymbol{\mu}))^2 \geq 0 \quad \text{with} \quad \boldsymbol{v} \in \mathbb{R}^d, \tag{B7}$$

but the covariance matrix calculated from our ensemble is occasionally indefinite. The reason for the covariance matrix being indefinite is a numerical error in the calculation of this matrix. In fact, the calculation of the eigenvalues $\lambda$ results in negative values in the order of $\mathcal{O}(10^{-20})$.

For this purpose, the identity matrix $\mathbf{I}$, which is multiplied by a scalar $\lambda_{\max}$, is added to the covariance matrix. The value of $\lambda_{\max}$ is in the order of magnitude of the largest negative eigenvalue of $\mathbf{Q}$. Thus, the regularized covariance matrix reads

$$\mathbf{Q}_{\text{Reg.}} = \mathbf{Q} + \lambda_{\max}\mathbf{I}. \tag{B8}$$

In our experiments, the smallest variance on the main diagonal of the covariance matrix is still 16 orders of magnitude larger than $\lambda_{\max}$ such that the influence of this correction is negligible and does not change the results.

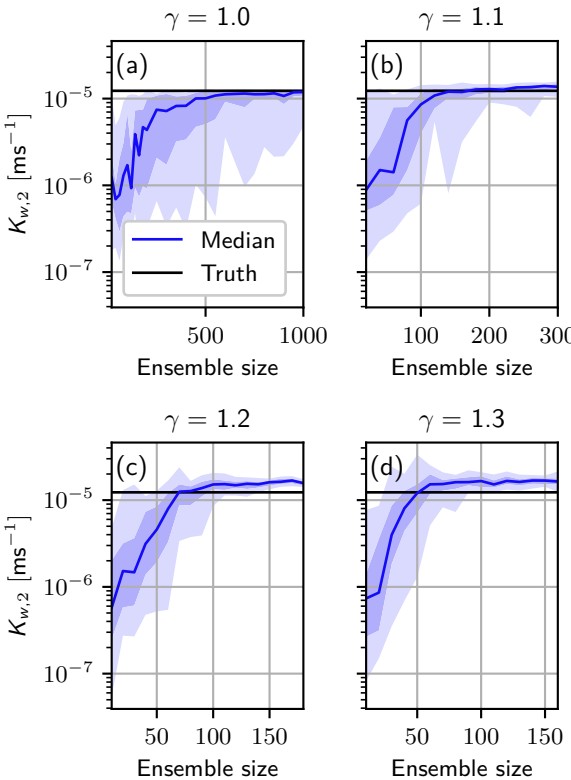

**Figure C1.** The mean saturated conductivity in the second layer after the data assimilation run for 40 different seeds and for varying factors of $\gamma_p$ (Eq. (14)) (a): $\gamma_p = 1.0$, (b): $\gamma_p = 1.1$, (c): $\gamma_p = 1.2$ and (d): $\gamma_p = 1.3$. The blue areas represent the 70 %-quantile (darker blue) and the 90 %-quantile (light blue), respectively. Note the different scaling of the x-axes.

## Appendix C: Dependence of $K_{w,2}$ on the tuning parameter $\gamma$

The saturated conductivity in the second layer is analyzed in the same setup as in Sect. 6.1. The assimilation is run for 40 different seeds, varying ensemble sizes and for four different tuning parameters $\gamma$ (see Eq. (14)). The remaining setup of the system is identical as in Sect. 4.

5    Figure C1 shows the mean saturated conductivity in the second layer $K_{w,2}$ after the data assimilation run, including the 70 %-quantile (darker blue area) and the 90 %-quantile (light blue area) of the 40 runs with different seeds.

For all ensemble sizes, the filter either degenerates or finds the true value. Increasing the ensemble size increases the number of successful runs. The degeneration of the filter can directly be seen in the effective sample size, which drops to $N_{\mathrm{eff}} = 1$. Therefore, we emphasize the need to control whether the filter degenerates or not, to ensure a meaningful result. Results

10   generated with a degenerated filter must not be used.

For the case $\gamma_p = 1.0$ (see Fig. C1a), which does not change the covariance matrix, the filter needs about 800 ensemble members to converge for 70 % of the seeds. It still degenerates for some seeds. Increasing the tuning factor for the parameter to $\gamma_p = 1.1$ (see Fig. C1b), reduces the necessary ensemble size and the seed dependency. For 300 ensemble members, the 90 %-quantile converges to the truth.

5      In Fig. C1c the tuning parameter is equal to the one used in the case study in Sect. 4. For less than 100 ensemble members, the behavior of the filter is seed dependent. While for some seeds the filter still converges for 20 ensemble members, it degenerates in most of them. For 100 ensemble members, the ensemble size used in the case study, the filter converges for every of the 40 seeds.

     The apparent bias to a larger saturated conductivity for $\gamma_p = 1.2$ is compensated by the other two estimated parameters in 10   this layer, such that the conductivity function Eq. (19) is almost identical to the truth in the measured water content range.

     Increasing the factor to $\gamma_p = 1.3$ (see Fig. C1d), does not change the result significantly compared to the case $\gamma_p = 1.2$ (C1c). However, choosing a too large value for the tuning parameter results in an increasing uncertainty, which leads to a divergent ensemble for insensitive parameters like $\alpha_1$. Therefore, it is important to check the results and adjust the tuning parameter accordingly. It is always possible to increase the ensemble size and run the assimilation without using the parameter 15   $\gamma$. The behavior of $\alpha_1$ and the remaining parameters can be found in the Supplementary.

*Data availability.* The data used for the figures is available online on heiDATA (Berg et al., 2019).

*Competing interests.* All authors declare that they have no competing interests.

*Acknowledgements.* We thank the editor Harrie-Jan Hendricks Franssen and the two reviewers Damiano Pasetto and Jasper Vrugt for their comments, which helped to improve this paper.

20      This research is funded by Deutsche Forschungsgemeinschaft (DFG) through project RO 1080/12-1. Hannes H. Bauser and Daniel Berg were supported in part by the Heidelberg Graduate School of Mathematical and Computational Methods for the Sciences (HGS MathComp), funded by DFG grant GSC 220 in the German Universities Excellence Initiative.

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
