# Peer review of "Covariance resampling for particle filter – state and parameter estimation for soil hydrology"

_Hydrology and Earth System Sciences, 2018_

## Referee Comment (RC1) · Anonymous Referee #1 · 7 May 2018

The paper introduces a new resampling method for particle filters, well suited to estimate both state variables and model parameters in a sequential DA approach. The well-known Universal Resampling Approach is modified by assigning new weights to the particles that should be duplicated, without actually duplicating the particles. These weights are proportional to the number of times the particles are selected in the universal resampling. To keep constant the ensemble size, new particles (states and parameters) are then generated by sampling from a multivariate Gaussian distribution having the same mean and covariance of the weighted particles. To avoid the degeneracy of the filter, the covariance is inflated using a multiplicative factor.

The proposed method is applied to a synthetic 1-d infiltration problem in a porous media constituted of two layers. Initial conditions and soil parameters (saturated hydraulic

conductivity and two parameters of the Van-Genuchten equations, for both layers) are considered uncertain. The authors demonstrate that, in the considered example, the proposed pf well retrieves the state variables of the system and the soil parameters.

The paper is well written and the method is clear. At my knowledge, the proposed resampling technique is new and I really like it, since it gives the possibility to propagate realizations consistent with the model equations (a limitation of EnKF – see e.g., Pasetto et al. 2012) and, at the same time, the possibility of sampling new particles, which is fundamental to explore the parameter space.

The paper should be considered for publication in HESS since the topic is relevant to the hydrological community. However, there are some limitations that need to be considered, especially taking into account that the method is new and the results are presented for just a single application:

1) The paper does not present any result on the convergence of the filter with respect to the ensemble size: it would be important to show that at least the first and second moments of state and parameters converge toward the correct solution when N increases, and that the results are insensitive to the particular seed used. This analysis would also help justifying the choice of N=100.

2) In a similar way, the sensitivity of the filter to the multiplicative factor gamma for the parameters (selected to be 1.2) should be presented. Could the authors give an advice to the readers on how to choose gamma for a different problem?

3) A comparison of the results against the SIR using universal resampling (or the methodology proposed in Moradkhani et al. 2005) would help to understand if the proposed PF is retrieving the correct solution (in terms of both mean and covariance) and which are the practical advantages of the proposed resampling step.

With the addition of these results, this work would become an outstanding paper. Since these analyses are time-consuming, I am proposing major revisions to let the authors

more time for the required computations.

MINOR COMMENTS

Please revise the numbers and labels on the x and y axis for all figures. Probably there was an error with the software used to produce the figures.

L8, p1: ' With just 100 particles'. In large scale applications 100 particles are frequently adopted. However, in this 1-d scenario it is difficult to assess is 100 particles are 'small', especially without presenting a comparison with other approaches and/or the sensitivity of the results to the number of particles.

L8-10, p1: 'The estimated states and parameters are tested with a free run after the assimilation, which is found to be in good agreement with the synthetic truth'. PFs (and DA in general) are meant to assess not only the mean value of states and parameters, but also their covariance (if not all the pdf). To assess if the covariance computed is correct, a comparison with respect to other DA schemes would be required.

L20, p1: 'The EnKF based on Richards equation..'. EnKF is applied to Richards equation, not 'based on'. Please rephrase.

L23, p2: what does it mean 'without additional model evaluation'? This statement would be more relevant if the results of the proposed method are compared against more traditional PFs (e.g. SIR with a standard resampling).

L27,p2: missing point.

Section 3 and part of section 2: particle filters refer to a broad class of methods (see, e.g., Arulampalam 2002). The authors are mainly describing the Sequential Importance Resampling technique. Please clarify this point in the paper, so that readers familiar with PFs can easily understand which technique has been modified.

Fig1: Write evapotranspiration instead of evaporation.

Eq. 12, Page 5. I was expecting to see the weight $w_i$ in the summation. Please,

provide a reference for the Bessel correction.

Lines 5-10, p9: it is not clear how the covariance matrix for the initial ensemble has been generated. Which matrices are multiplied in step 2? Is step one performed after step 2, to ensure zero correlation of the states across the two layers?

Page 13,L1: add 'of' after estimate

——————

REFERENCES

Arulampalam MS, Maskell S, Gordon N, Clapp T. A tutorial on particle filters for online nonlinear/non-Gaussian Bayesian tracking. IEEE T. Signal Proces. 2002;50(2):174–88.

Moradkhani H, Hsu K-L, Gupta H., and Sorooshian, S.: Uncertainty assessment of hydrologic model states and parameters: Sequential data assimilation using the particle filter. Water Resour. Res., 2005; 41.

Pasetto D, Camporese M, Putti M. Ensemble Kalman filter versus particle filter for a physically-based coupled surface-subsurface model. Adv. Water Resour., 2012; 47(1), 1-13.

---

## Referee Comment (RC2) · J. Vrugt (Referee) · 26 Jun 2018

Particle filters (PFs) have found widespread application and use for state and/or parameter estimation of dynamic system models. The premise of such filters is that they provide an exact approximation of the state forecast distribution. Yet, particle filters are not necessarily efficient as they may require a very large number of ensemble members (so-called particles) to approximate closely the evolving state distribution. This is particularly true in high-dimensional state spaces and complicates significantly the practical and/or real-time application of particle filters. What is more, particle filters are prone to sample impoverishment, that is, after a number of so-called assimilation steps, a very large number of the particles receives a negligible weight. These particles thus contribute little to the state forecast distribution and should be discarded/eliminated to

a) refocus the thrust of the filter on the high-density region of the state space, and b) maintain an adequate filter efficiency and use of CPU resources. In the past decades, different resampling methods have been proposed and/or used to periodically rejuvenate the particle ensemble and ensure an adequate tracking of the evolving state distribution. Of these, Sequential Importance Resampling (SIR) has found most application and use. This method re-samples the particle ensemble using the computed weights of the $N$ particles. These weights are simply equivalent to the product of the prior and the likelihood of the particle's simulated trajectory. Whereas this SIR method is computationally efficient, it typically leads to a resampled ensemble with many copies of only a few of the "best" members (those with highest weights). In theory, this should not necessarily be a problem as the model operator (transition density) would disperse identical copies of the initial states as a result of the stochastic model error. This would work well in practice if the transition density of the state vector closely approximates the underlying system behavior. Unfortunately, even a modest deviation of the model operator from the actual system dynamics would deteriorate the particle ensemble to a point that most particles receive only a negligible weight. Thus, resampling is of crucial importance to periodically rejuvenate the particle ensemble and make sure that the simulated state PDF mimics closely the observed system behavior. Note that Ensemble Kalman filters do not suffer this same problem with sample impoverishment as they use a state analysis step to update the state forecasts of the $N$ ensemble members each time an observation is becoming available.

In this paper, the authors present a new resampling method to improve the efficiency and practical application of particle filters. This resampling method stores only a single copy of the $M$ "best" particles determined with a standard resampling method, say SIR, and simulates the $N - M$ "open spots" by drawing from a $m-variate normal distribution with mean and covariance matrix derived from the m-dimensional state forecast distribution.$

This paper considers an important practical (and theoretical) problem in hydrologic data assimilation. This topic is relevant to HESS and should be of interest as well to

an audience outside hydrology as it involves improvements to an existing method. The paper is generally well written but would benefit from careful editing. I now list my main comments.

1. The authors should consider a more realistic or appealing case study. Indeed, the present one-dimensional Richards' type flow problem with two horizontal layers is too simple to really demonstrate the advantages of the proposed covariance resampling methodology. The authors should consider a range of different state dimensionalities, $m$, to demonstrate that their method does not suffer from particle impoverishment. The authors should consider the Lorenz96 model with $m = 40$ state variables - this would demonstrate (or not) that the proposed resampling method works well in higher dimensional state spaces and would track closely the observed system dynamics. Such case study would make the paper much stronger and more appealing to those interested in methodological developments.

2. The authors refer to Figure 2 for a demonstration of the proposed covariance resampling method. I do not necessarily find this illustration to be particularly informative - that is - I think the authors can do a better job in detailing the proposed resampling method. The present animation assumes as if the target distribution is already well described with the present forecast distribution. In practice, this is often not true, certainly in higher dimensional state spaces. Also, I think the authors should differentiate between the state forecast density and the "true" or "unobserved" state forecast PDF. Then detail how the resampling works in practice. Personally, I always enjoy reading well-crafted algorithmic recipes (and associated coding) as those detail a step-by-step plan of how to implement the steps detailed in the main text.

3. On Page 5 (top part) the authors list some previous approaches that have been used to resample the particle ensemble. I think the authors should mention whether each of these listed approaches leave the target state PDF invariant - that is - they lead to an exact approximation of the evolving target PDF. This may not necessarily be of concern to most hydrologists but is a requirement for methods to find widespread

application and use. The same comment applies to the Introduction Section of the paper. In other words, I think it is good to emphasize that ad-hoc methods may provide results - but that such methods may not enjoy statistical underpinning.

4. I think the paper will be better if the authors replace Equation (1) with a recursive implementation of Bayes Law. This will make clear the relationship between the prior and posterior state PDF, how Equation (8) ties into this, and defines the importance weight, incremental importance weight and normalized importance weight. As it stands the current theory section omits completely the dimension of time - and this is key to state estimation.

5. I think it may be worthwhile to tie Equation (2) to the marginal likelihood. This is what you want to maximize with parameter estimation - but is of no real concern/importance for state estimation.

6. Do not understand the need for Equation (6) - and also do not necessarily directly understand how the normalized weights lead to the normalization constant. This denominator, or evidence, does not require the importance weights to add up to unity, right?

7. The authors assign a weight of 1/N to the samples drawn from the $m$-variate normal distribution. I am not sure whether this leaves the state PDF invariant. The authors treat as if the samples from the multivariate normal have an equal weight - this is fine if ALL $N$ samples were drawn from the multivariate normal PDF - but the state vectors drawn from this normal PDF are combined with the existing M "best" particles of the state forecast distribution - and those latter ones do not have a weight of 1/N. This cannot be justified theoretically. So, it is of crucial importance to demonstrate that the proposed resampling method leads to the exact target PDF.

8. The present resampling method relies heavily on the simulated state forecast distribution. If this distribution does not properly approximate the actual target PDF then resampling will provide $N$ unique samples but those state vectors are not expected

to produce a proper forecast PDF at the next time when a subsequent measurement becomes available. In other words, the present resampling method assumes that the transition density (model operator) approximates closely the true system dynamics. Once the state forecast PDF is systematically biased (likely to happen in real-world application) then the present resampling method may not necessarily enhance particle filtering results.

9. The authors use a perturbation factor, gamma, to inflate or deflate the covariance matrix of the normal resampling PDF. There is no justification for this - that is - its value is entirely subjective - indeed, one can tune gamma to provide appealing results, yet the value of gamma should guarantee an exact approximation of the target PDF (see Vrugt et al., 2013).

10. The authors use state augmentation to estimate jointly the model's state variables and parameter values. Do I conclude correctly that the authors use the normal resampling PDF to generate new parameter vectors? So, the resampling method assumes that the state variables and parameter values are multivariate normal. Would it not make sense to implement a mixture distribution instead - and estimate this distribution from the forecasted state/parameter distribution of the $N$ particles?

11. The synthetic case study presented in the paper satisfies the assumption of a perfect model and thus transition density; in other words, the presented resampling method should work well as the forecast PDF (state/parameter) is not expected to deviate systematically from the observed data and system behavior. A real-world case study with actual measured data would provide a much stronger test of the proposed method. For example, one can use the Lorenz96 model to create an artificial data set - and then use an alternative model formulation to test, evaluate and benchmark the proposed covariance resampling method.

For now, I'll leave it with these comments. In summary, I think the authors have to investigate in more depth the statistical rigor of their resampling methodology. I have

serious concerns about the statistical validity of the proposed method - as far as I understand the details I do not think that the proposed resampling method leads to an exact approximation of the evolving target PDF. What is more, the authors should consider a more complex case study involving a higher dimensional state space and a transition density (model operator) that cannot track exactly observed system dynamics. If the authors can show that their method works well with a relatively inferior model then this would demonstrate in strongest possible terms the advantages of the proposed resampling method. Indeed, it is then when particle filters break down. Also, the authors should provide theoretical justification for the use of a multiplicative factor to scale the covariance matrix of the multivariate normal resampling PDF. This factor is entirely subjective and can be tuned so that one achieves desired results - but what is the value of this scalar for other studies? How do we know what value for gamma to take in practice?

I do not want to discourage the authors, but I believe that proper resampling necessitates the use of MCMC simulation and re-simulation of part of the historic trajectory to determine whether to accept a proposal or not. Such re-simulation is not particularly appealing, yet required to track properly the evolving state PDF. I elude to the work we published in Vrugt et al. (2013) (particle filtering with DREAM resampling/resimulation) - which ultimately led me to conclude that particle filters are not particularly useful for real-world application to complex systems - unless you have at your disposal sufficient CPU resources and can afford the use of an excessively large particle ensemble. This may then guarantee a sufficient coverage of the state space so that particle resampling can rapidly rectify systematic deviations between the forecast PDF and the "measured" state PDF as expressed in the measured data.

I hope my comments are useful to further improve this paper. As always, my comments/interpretations may be wrong! As usual, I welcome dialogue with the authors on this and/or related topics. Jasper A. Vrugt Irvine, June 26, 2018

---

## Author Comment (AC1) · 15 Aug 2018

**Summary:** *The paper introduces a new resampling method for particle filters, well suited to estimate both state variables and model parameters in a sequential DA approach. The well-known Universal Resampling Approach is modified by assigning new weights to the particles that should be duplicated, without actually duplicating the particles. These weights are proportional to the number of times the particles are selected in the universal resampling. To keep constant the ensemble size, new particles (states and parameters) are then generated by sampling from a multivariate Gaussian distribution having the same mean and covariance of the weighted particles. To avoid the degeneracy of the filter, the covariance is inflated using a multiplicative factor.*
*The proposed method is applied to a synthetic 1-d infiltration problem in a porous media constituted of two layers. Initial conditions and soil parameters (saturated hydraulic conductivity and two parameters of the Van-Genuchten equations, for both layers) are considered uncertain. The authors demonstrate that, in the considered example, the proposed pf well retrieves the state variables of the system and the soil parameters.*
*The paper is well written and the method is clear. At my knowledge, the proposed resampling technique is new and I really like it, since it gives the possibility to propagate realizations consistent with the model equations (a limitation of EnKF – see e.g., Pasetto et al. 2012) and, at the same time, the possibility of sampling new particles, which is fundamental to explore the parameter space.*

**Reply:** We thank the reviewer for the detailed comments and suggestions, which will help to improve our manuscript. In the following we provide the answers to the comments.

**Specific comments**

**Comment:** *The paper does not present any result on the convergence of the filter with respect to the ensemble size: it would be important to show that at least the first and second moments of state and parameters converge toward the correct solution when N increases, and that the results are insensitive to the particular seed used. This analysis would also help justifying the choice of N=100.*

**Reply:** Thank you for pointing this out. A continuous convergence to the truth with an increasing ensemble size does not occur. For varying seeds combined with a small fixed ensemble size (e.g. 20) the filter converges to

the truth or degenerates for different seeds. Increasing the ensemble size leads to less cases that degenerate and ultimately the filter converges to the true value for every chosen seed. The resulting parameters are in a narrow range around the truth except for the insensitive parameter $\alpha_1$. A paragraph about the seed and ensemble size dependence of the resulting parameters will be added to clarify the behaviour of the filter.

**Comment:** *In a similar way, the sensitivity of the filter to the multiplicative factor gamma for the parameters (selected to be 1.2) should be presented. Could the authors give an advice to the readers on how to choose gamma for a different problem?*

**Reply:** The multiplicative factor is a tuning factor that depends on the specific problem and can be used to increase the efficiency of the filter. The covariance resampling also works with a neutral factor of 1.0 but needs (in this case) approximately an order of magnitude more ensemble members for convergence. The factor will be discussed in more detail in the course of the convergence analysis.

**Comment:** *A comparison of the results against the SIR using universal resampling (or the methodology proposed in Moradkhani et al. 2005) would help to understand if the proposed PF is retrieving the correct solution (in terms of both mean and covariance) and which are the practical advantages of the proposed resampling step.*

**Reply:** Using the SIR with universal resampling leads to filter degeneration after a few assimilation cycles because the model equation in our case study does not have a stochastic model error. Therefore, after resampling, the duplicated particles will stay identical after forward propagation which leads to filter degeneration. For the resampling techniques of Moradkhani et al. (2005), we tested different tuning factors in the interval $[0.2, 0.8]$ for the parameter resampling. The tuning factor modifies the variance of the perturbation. We also tried different ensemble sizes in the interval $[100, 1000]$. Using the same initial conditions as in the presented case, we were not able to achieve converging results. Changing the state's initial condition from the interpolation to the truth, the resampling of Moradkhani et al. (2005) was able to follow the truth in state space for 400 ensemble members and a factor of 0.6, but without convergence in parameter space and the ensemble of parameter $\alpha_1$ diverges.

**Minor comments**

**Comment:** *Please revise the numbers and labels on the x and y axis for all figures. Probably there was an error with the software used to produce the figures.*
**Reply:** Thank you very much, we will revise the figures. This is an issue with the used font during the plotting, which is not correctly displayed after uploading the .pdf file. We are in contact with Copernicus to solve it in the revised version.

**Comment:** *L8, p1: 'With just 100 particles'. In large scale applications 100 particles are frequently adopted. However, in this 1-d scenario it is difficult to assess is 100 particles are 'small', especially without presenting a comparison with other approaches and/or the sensitivity of the results to the number of particles.*
**Reply:** You are right. The comparison to other approaches is difficult because the case studies have a different setup, therefore we will delete the 'just'.

**Comment:** *L8-10, p1: 'The estimated states and parameters are tested with a free run after the assimilation, which is found to be in good agreement with the synthetic truth'. PFs (and DA in general) are meant to assess not only the mean value of states and parameters, but also their covariance (if not all the pdf). To assess if the covariance computed is correct, a comparison with respect to other DA schemes would be required.*
**Reply:** The entire ensemble is propagated forward in time and the ensemble mean is calculated using the weights at the last assimilated observation. This way we consider the mean of the propagated distribution. We will clarify this point.

**Comment:** *L20, p1: 'The EnKF based on Richards equation..'. EnKF is applied to Richards equation, not 'based on'. Please rephrase.*
**Reply:** Thank you for pointing this out. We will rephrase the sentence.

**Comment:** *L23, p2: what does it mean 'without additional model evaluation'? This statement would be more relevant if the results of the proposed method are compared against more traditional PFs (e.g. SIR with a standard resampling).*
**Reply:** In this case we considered the MCMC resampling, which needs additional model runs for the resampling process. These runs are usually expensive and therefore, we think the absence of additional model runs as an advantage. We will clarify the statement.

**Comment:** *L27,p2: missing point.*
**Reply:** Thank you for noticing the missing point. We will correct it.

**Comment:** *Section 3 and part of section 2: particle filters refer to a broad class of methods (see, e.g., Arulampalam 2002). The authors are mainly describing the Sequential Importance Resampling technique. Please clarify this point in the paper, so that readers familiar with PFs can easily understand which technique has been modified.*
**Reply:** We will clarify this point.

**Comment:** *Fig1: Write evapotranspiration instead of evaporation.*
**Reply:** In our simulations we do not have plants and therefore, no transpiration.

**Comment:** *Eq. 12, Page 5. I was expecting to see the weight $w_i$ in the summation. Please, provide a reference for the Bessel correction.*
**Reply:** You are right with expecting the weight $w_i$ in the summation. The missing weight in the equation is a typing error. The correct equation is:

$$\mathbf{P}^{\mathrm{f}} = \frac{1}{1 - \sum_{i=1}^{N} w_i^2} \sum_{i=1}^{N} w_i \left[\boldsymbol{u}_i - \overline{\boldsymbol{u}}\right] \left[\boldsymbol{u}_i - \overline{\boldsymbol{u}}\right]^{\intercal}. \tag{1}$$

It is possible to represent the weights using a larger number of equal weighted particles. For example, a particle with $w_i = 1/2$, $u_i = 1$ and two particles with $w_i = 1/4$, $u_i = 2$ is equal to having two particle with $w_i = 1/4$, $u_i = 1$ and two particles with $w_i = 1/4$, $u_i = 2$.

Inserting equal weighted particles in this equation results in

$$\mathbf{P}^{\mathrm{f}} = \frac{1}{1 - \frac{1}{N}} \sum_{i=1}^{N} \frac{1}{N} \left[\boldsymbol{u}_i - \overline{\boldsymbol{u}}\right] \left[\boldsymbol{u}_i - \overline{\boldsymbol{u}}\right]^{\intercal} \tag{2}$$

$$= \frac{1}{N-1} \sum_{i=1}^{N} \left[\boldsymbol{u}_i - \overline{\boldsymbol{u}}\right] \left[\boldsymbol{u}_i - \overline{\boldsymbol{u}}\right]^{\intercal}. \tag{3}$$

This is equal to the correction of an unbiased estimate of the covariance.

**Comment:** *Lines 5-10, p9: it is not clear how the covariance matrix for the initial ensemble has been generated. Which matrices are multiplied in step 2? Is step one performed after step 2, to ensure zero correlation of the states across the two layers?*
**Reply:** Thank you for pointing this out. Every entry of the initial covariance matrix is set to $0.003^2$. In the first step, covariances across the layer boundary are set to zero. The Gaspari and Cohn function is multiplied componentwise (not written in the manuscript) with the resulting matrix. Therefore,

both steps are exchangeable. The component-wise multiplication with the Gaspari and Cohn function results in a covariance that decreases with the distance. We will rephrase and extend this paragraph to clarify the steps.

**Comment:** *Page 13, L1: add 'of' after estimate*
**Reply:**  We will add the 'of'.

**References**

Hamid Moradkhani, Kuo-Lin Hsu, Hoshin Gupta, and Soroosh Sorooshian. Uncertainty assessment of hydrologic model states and parameters: Sequential data assimilation using the particle filter. *Water Resources Research*, 41(5), 2005. doi:10.1029/2004WR003604. URL `https://agupubs.onlinelibrary.wiley.com/doi/abs/10.1029/2004WR003604`.

---

## Author Comment (AC2) · 15 Aug 2018

**Summary:** *Particle filters (PFs) have found widespread application and use for state and/or parameter estimation of dynamic system models. The premise of such filters is that they provide an exact approximation of the state forecast distribution. Yet, particle filters are not necessarily efficient as they may require a very large number of ensemble members (so-called particles) to approximate closely the evolving state distribution. This is particularly true in high-dimensional state spaces and complicates significantly the practical and/or real-time application of particle filters. What is more, particle filters are prone to sample impoverishment, that is, after a number of so-called assimilation steps, a very large number of the particles receives a negligible weight. These particles thus contribute little to the state forecast distribution and should be discarded/eliminated to a) refocus the thrust of the filter on the high-density region of the state space, and b) maintain an adequate filter efficiency and use of CPU resources. In the past decades, different resampling methods have been proposed and/or used to periodically rejuvenate the particle ensemble and ensure an adequate tracking of the evolving state distribution. Of these, Sequential Importance Resampling (SIR) has found most application and use. This method re-samples the particle ensemble using the computed weights of the N particles. These weights are simply equivalent to the product of the prior and the likelihood of the particle?s simulated trajectory. Whereas this SIR method is computationally efficient, it typically leads to a resampled ensemble with many copies of only a few of the "best" members (those with highest weights). In theory, this should not necessarily be a problem as the model operator (transition density) would disperse identical copies of the initial states as a result of the stochastic model error. This would work well in practice if the transition density of the state vector closely approximates the underlying system behavior. Unfortunately, even a modest deviation of the model operator from the actual system dynamics would deteriorate the particle ensemble to a point that most particles receive only a negligible weight. Thus, resampling is of crucial importance to periodically rejuvenate the particle ensemble and make sure that the simulated state PDF mimics closely the observed system behavior. Note that Ensemble Kalman filters do not suffer this same problem with sample impoverishment as they use a state analysis step to update the state forecasts of the N ensemble members each time an observation is becoming available.*

*In this paper, the authors present a new resampling method to improve the efficiency and practical application of particle filters. This resampling method stores only a single copy of the M "best" particles determined with a standard resampling method, say SIR, and simulates the N – M "open spots" by drawing from a m-variate normal distribution with mean and covariance from the m-dimension.*

**Reply:** We thank Jasper Vrugt for the detailed comments and suggestions, which will help to improve our manuscript. We also welcome a dialogue on this and related topics. In the following we provide our answers to Jaspar Vrugt's comments.

**Specific comments**

**Comment:** *1. The authors should consider a more realistic or appealing case study. Indeed, the present one-dimensional Richards' type flow problem with two horizontal layers is too simple to really demonstrate the advantages of the proposed covariance resampling methodology. The authors should consider a range of different state dimensionalities, m, to demonstrate that their method does not suffer from particle impoverishment. The authors should consider the Lorenz96 model with m = 40 state variables - this would demonstrate (or not) that the proposed resampling method works well in higher dimensional state spaces and would track closely the observed system dynamics. Such case study would make the paper much stronger and more appealing to those interested in methodological developments.*
**Reply:** Thank you for your comment. We propose the covariance resampling for state and parameter estimation in soil hydrology. This resampling aims to tackle the challenges of absent or unknown model errors for parameter estimation. Without perturbation particles will stay identical in the absence of a model error and the filter will degenerate. In the presented case study, we showed the effectiveness of the method for a soil hydrological problem. The Lorenz-96 model is an artificial atmospheric model and the behaviour differs significantly from soil hydrology using Richards' equation. Because of our focus on soil hydrology, the Lorenz-96 model is not used.

Just like other particle filters using only resampling, the proposed filter will also suffer from filter impoverishment in high-dimensional systems. The covariance resampling does not aim to lift the 'curse of dimensionality'.

**Comment:** *2. The authors refer to Figure 2 for a demonstration of the proposed covariance resampling method. I do not necessarily find this illustration to be particularly informative - that is - I think the authors can do a better job in detailing the proposed resampling method. The present animation assumes as if the target distribution is already well described with the present forecast distribution. In practice, this is often not true, certainly in higher dimensional state spaces. Also, I think the authors should differentiate between the state forecast density and the "true" or "unobserved" state fore-*

*cast PDF. Then detail how the resampling works in practice. Personally, I always enjoy reading well-crafted algorithmic recipes (and associated coding) as those detail a step-by-step plan of how to implement the steps detailed in the main text.*

**Reply:** Thank you for this suggestion. We agree that pseudo code is a more detailed recipe for implementing the actual algorithm. We think that a picture is more illustrative. Therefore, we will add the pseudo code in the appendix as an additional information for the reader.

**Comment:** *3. On Page 5 (top part) the authors list some previous approaches that have been used to resample the particle ensemble. I think the authors should mention whether each of these listed approaches leave the target state PDF invariant - that is - they lead to an exact approximation of the evolving target PDF. This may not necessarily be of concern to most hydrologists but is a requirement for methods to find widespread application and use. The same comment applies to the Introduction Section of the paper. In other words, I think it is good to emphasize that ad-hoc methods may provide results - but that such methods may not enjoy statistical underpinning.*

**Reply:** All of the mentioned methods approximate the posterior PDF. Of course some approximations are closer to the 'true' PDF than others. As long as all PDFs/errors are described correctly, we agree that e.g. the MCMC-resampling presented in Vrugt et al. (2013) is one of the methods that is closer to this target PDF. However, the model errors e.g. unrepresented physics, are typically unknown. In this case, the estimated target PDF is not the 'true' PDF anymore. Even without resampling and in the limit of infinite ensemble members the estimated PDF will be erroneous. Then it can be an advantage dropping the rigorous formalism to gain stability for the method. We will clarify this.

**Comment:** *4. I think the paper will be better if the authors replace Equation (1) with a recursive implementation of Bayes Law. This will make clear the relationship between the prior and posterior state PDF, how Equation (8) ties into this, and defines the importance weight, incremental importance weight and normalized importance weight. As it stands the current theory section omits completely the dimension of time - and this is key to state estimation.*

**Reply:** Thank you for this suggestion. We described the update at one time step. We agree that this is incomplete. We will extend the description, add a time index and use the recursive Bayes filter equation to clarify the update and analysis steps.

**Comment:** *5. I think it may be worthwhile to tie Equation (2) to the marginal likelihood. This is what you want to maximize with parameter esti-*

*mation - but is of no real concern/importance for state estimation.*

**Reply:** Equation (2) is the marginal likelihood of the observation and is a normalisation constant in Bayes' theorem. This normalisation constant is calculated using Equation (6) (see also next reply).

**Comment:** *6. Do not understand the need for Equation (6) - and also do not necessarily directly understand how the normalized weights lead to the normalization constant. This denominator, or evidence, does not require the importance weights to add up to unity, right?*

**Reply:** Thank you for the comment. We added this equation to clarify the calculation of the factor $P(d)$. Since this factor is a normalisation constant, which is the same for all weights, it is possible to calculate this factor from the normalisation of the weights. Therefore, to calculate this factor, it is necessary that the importance weights add up to unity. We think adding the recursive formulation of Bayes' theorem, like you suggested in comment 4, will clarify this. The equations will then change to

$$w_i^k = w_i^{k-1} \frac{P(\boldsymbol{d}^k | \boldsymbol{u}_i^k)}{P(\boldsymbol{d}^k)} \tag{1}$$

and

$$\sum_{i=0}^{N} w_i^k \overset{!}{=} 1 \quad \Rightarrow \quad P(\boldsymbol{d}^k) = \sum_{i=0}^{N} w_i^{k-1} P(\boldsymbol{d}^k | \boldsymbol{u}_i^k) \, , \tag{2}$$

where $k$ is the discrete time index.

**Comment:** *7. The authors assign a weight of 1/N to the samples drawn from the m-variate normal distribution. I am not sure whether this leaves the state PDF invariant. The authors treat as if the samples from the multivariate normal have an equal weight - this is fine if ALL N samples were drawn from the multivariate normal PDF - but the state vectors drawn from this normal PDF are combined with the existing M "best" particles of the state forecast distribution - and those latter ones do not have a weight of 1/N. This cannot be justified theoretically. So, it is of crucial importance to demonstrate that the proposed resampling method leads to the exact target PDF.*

**Reply:** Thank you for the comment. This is correct, the new particles do not leave the PDF invariant. The retained particles do not have the weights of $\frac{1}{N}$. The weights are changed such that the resampling is similar to the universal resampling, except that the ensemble size is reduced. At this point the PDF is invariant in the limit of large ensemble sizes. To increase the ensemble size to $N$ again, new particles are drawn from the multivariate

normal distribution. This alters the estimated posterior distribution. The mean and variance of the distribution are invariant under this process. For a large effective sample size, the overall structure remains close to the original posterior PDF. For small effective sample sizes, a large fraction of the particles is resampled such that the posterior distribution is dominated by the approximated multivariate Gaussian. We will extend the description of the method to clarify when the distribution is altered.

**Comment:** *8. The present resampling method relies heavily on the simulated state forecast distribution. If this distribution does not properly approximate the actual target PDF then resampling will provide N unique samples but those state vectors are not expected to produce a proper forecast PDF at the next time when a subsequent measurement becomes available. In other words, the present resampling method assumes that the transition density (model operator) approximates closely the true system dynamics. Once the state forecast PDF is systematically biased (likely to happen in real-world application) then the present resampling method may not necessarily enhance particle filtering results.*
**Reply:** Thank you for your input. You are right with the conclusion that a systematically bias will provide a challenge, if this bias is not represented in the model equation. However, in this case also an accurate representation of the PDF will result in a wrong forecast PDF for the next time step. In such a case an ensemble of unique samples has probably a better chance to not degenerate in the next time step because they can explorer the state space more widely.

**Comment:** *9. The authors use a perturbation factor, gamma, to inflate or deflate the covariance matrix of the normal resampling PDF. There is no justification for this - that is - its value is entirely subjective - indeed, one can tune gamma to provide appealing results, yet the value of gamma should guarantee an exact approximation of the target PDF (see Vrugt et al., 2013).*
**Reply:** It is correct that the multiplicative factor $\gamma$ is a tuning parameter to inflate or deflate the covariance matrix. This factor can be used to increase the efficiency of the method. Such tuning parameters are common in different data assimilation methods. The factor can be set to $\gamma = 1.0$, which also gives a converging result, however, the necessary particles are about one order of magnitude larger in our example. The reason for the necessity of more particles is that the parameters do not have their own dynamics. For small effective sample sizes, the variance can become very small such that the new particles do not deviate significantly. During the forward propagation the particles are not separated because of the missing dynamics and absent model error, which leads to possible filter degeneration. Using a factor larger

than 1 stabilises the filter for small effective sample sizes. We will add a paragraph about the convergence of the filter for different seeds and factors $\gamma$ with respect to the ensemble size.

**Comment:** *10. The authors use state augmentation to estimate jointly the model's state variables and parameter values. Do I conclude correctly that the authors use the normal resampling PDF to generate new parameter vectors? So, the resampling method assumes that the state variables and parameter values are multivariate normal. Would it not make sense to implement a mixture distribution instead - and estimate this distribution from the forecasted state/parameter distribution of the N particles?*
**Reply:** Your conclusion is correct, we approximate the posterior as a multivariate normal distribution for the generation of the new particles. A kernel density estimation would be possible but will neglect the correlation structure of the ensemble. A multivariate kernel density estimation would need too many particles to be practical. Therefore, using the multivariate normal distribution allows us to use the correlation information of the ensemble.

**Comment:** *11. The synthetic case study presented in the paper satisfies the assumption of a perfect model and thus transition density; in other words, the presented resampling method should work well as the forecast PDF (state/parameter) is not expected to deviate systematically from the observed data and system behavior. A real-world case study with actual measured data would provide a much stronger test of the proposed method. For example, one can use the Lorenz96 model to create an artificial data set - and then use an alternative model formulation to test, evaluate and benchmark the proposed covariance resampling method.*
**Reply:** We introduced this method as a possibility for state and parameter estimation using particle filters in soil hydrology. The paper is intended to present the concept of this method, which is shown in a synthetic example. Using real data has additional challenges, which would shift the focus of the paper away from the method towards these challenges.

**References**

Jasper A. Vrugt, Cajo J.F. ter Braak, Cees G.H. Diks, and Gerrit Schoups. Hydrologic data assimilation using particle Markov chain Monte Carlo simulation: Theory, concepts and applications. *Advances in Water Resources*, 51(Supplement C):457 – 478, 2013. ISSN 0309-1708. doi:10.1016/j.advwatres.2012.04.002. URL http://www.sciencedirect.

com/science/article/pii/S0309170812000863. 35th Year Anniversary Issue.

---

## Author Response (AR2)

**Answers to Damiano Pasetto**

**Summary:** *The Authors satisfactory replied to most of my comments. I appreciate that the Authors included the analysis on the filter convergence with respect the seed and the ensemble size, together with the sensitivity to the inflating parameter. This analysis strongly improved the paper.*

**Reply:** We thank Damiano Pasetto for the second review and appreciate his additional comments and suggestions, which will help to improve our manuscript. In the following we provide the answers to the comments.

**General comment**

**Comment:** *I am not sure if showing the specific convergence results on the saturated conductivity of the second layer (figure 10) is the best way to demonstrate the convergence. As first, because there is a small bias on this parameter (as discussed in P16, l 25). Thus it would be important to present the convergence results also on the other parameters, however this would require too many figures. As second, because the PF provides an approximation of the posterior distribution of the parameters, thus the analysis should take into account not only the mean of the estimated parameters, but also the ensemble spread, i.e. the covariance of the ensemble for that parameter. A possible alternative would be to present in the main manuscript the convergence of the error, for example the average RMSE on the water content during the free run and its ensemble spread. I suggest including the figures on the particular convergence on each parameter (such as figure 10) in the supplementary information.*

**Changes:** Thank you for your suggestion. We moved the section to the appendix (Line 1, Page 24 until Line 15, Page 25) and added the figures for the convergence of all parameters as supplementary information.
Line 5, Page 16 until Line 19, Page 18: We show now the RMSE of the water content and its variance at the last assimilation time.

**Detailed comments**

**Comment:** *P3, line 4: "Suppose a set?" the verb is missing.*
**Changes:** Thank you for pointing this out.
Line 4-10, Page 3: We changed the sentence to: "For a set of observations $\boldsymbol{d}^{1:k} = (\boldsymbol{d}^1, \boldsymbol{d}^2, \ldots, \boldsymbol{d}^{k-1}, \boldsymbol{d}^k)$, where the superscript denotes a discrete time index, the observations are assimilated sequentially using the recursive filter equation

$$P(\boldsymbol{u}^{0:k}|\boldsymbol{d}^{1:k}) = \frac{P(\boldsymbol{d}^{1:k}|\boldsymbol{u}^k)P(\boldsymbol{u}^k|\boldsymbol{d}^{1:k-1})}{P(\boldsymbol{d}^k)} \; ,$$

which follows from Bayes' theorem."

**Comment:** *P. 9, line 15: please delete on of the two "is" (repeated)*
**Changes:** Line 1, Page 10: We deleted one "is".

**Comment:** *P10, Eq 21 : using a pedix to the number to indicate the dimension is not clear, and I think it is not necessary to repeat this pedix in all the occurrences of these values. Why not using variables $\gamma_\theta$ and $\gamma_p$ (as in section 6.1), with the pedix just for this equation. Then, you can assign a numerical value directly to $\gamma_\theta$ and $\gamma_p$, without pedix. Please remove the pedix to the numerical values of $\gamma_\theta$ and $\gamma_p$.*
**Changes:** Thank you for your suggestion.
Line 5-7, Page 11 & throughout the manuscript: We changed the subscript accordingly.

**Comment:** *P. 16, line 1: in the section/subsections titles, sometimes the Authors use the upper case for all words, sometimes only for the first word. Please be consistent. Personally, I would use the upper case only for proper nouns.*
**Changes:** Thank you very much for pointing this out. We changed the titles accordingly.

**Comment:** *P. 16 Line 25: please replace "parameter" by "parameters"*
**Changes:** Line 9, Page 25: We replaced "parameter" by "parameters".

**Comment:** *P. 17, line 3: please replace "there" by "their". I suggest also to rephrase this sentence: model errors can have a structural or stochastic nature, different intensities, and the can manifest e.g. by bias in the results.*
**Changes:** Thank you for your suggestion.
Line 21-23, Page 18: We follow it and rephrased the sentence accordingly: "They can have a structural or stochastic nature, different intensities, and they can manifest e.g. by biases in the results."

**Comment:** *P17, line 7: please add "the": "In the course of this paper"*
**Changes:** Line 27, Page 18: We added "the".

**Comment:** *P18, lines 15/20: it is not clear which is the ensemble size used. It is 600/1200 with gamma =1 or 300/600 with gamma as in Eq 21?*
**Reply:** Thank you for pointing this out. We used 600/1200 with gamma as in Eq (21). For the 300/600 $\gamma$ is $\gamma_p = 1.2$ and $\gamma_\theta = 1.1$. These results are not shown.
**Changes:** Line 33, Page 18 until Line 4, Page 19: We clarified which ensemble size and $\gamma$ is used.

**Comment:** *Figure 7: to better appreciate the impact of the filter, is it possible to show also the water content associated to the initial values of the parameters (as in fig. 6?).*
**Changes:** We added the mean of an ensemble of free runs (Fig. 7, Page 14). The ensemble starts from the initial water content states (see Fig. 4), including their corresponding parameter sets and is propagated until the end of the data assimilation run. The ensemble members are not shown because it would strongly reduce the visibility of the actual results of the data assimilation (see Fig. AC1.1).

**Comment:** *Figure 9: Please specify if the RMSE is computed for each sample it the posterior parameter distribution, or only for the simulation associated with the mean of the posterior parameter distribution.*
**Reply:** The ensemble is propagated forward in time. The weights of the last assimilation time are used to calculate the mean of this forward run.
**Changes:** Line 8-9, Page 11 & Fig. 9, Page 16: We clarified this point.

**Comment:** *Figure 10: The PF method provides a distribution of the saturated conductivity for each seed and for each ensemble size. For this analysis, are the Authors considering the mean or median of that distribution or the whole distribution? If only the mean or median are considered, it would be important to provide also the convergence of the ensemble spread.*
**Reply:** We followed your suggestion in the general comment and show the ensemble spread and the RMSE of the water content (see also reply to the general comment). The parameter analysis is for the mean.
**Changes:** Fig. C1, Page 24 & Line 5, Page 24: We clarified this point and moved the section to Appendix C.

**Comment:** *Algorithm 1: line 1: at time $t_k$, the ensemble of state is $u_i^k$. Point (d). The initialization x=0 is missing at the beginning. The application of the tuning parameter is missing.*
**Reply:** The initialization is missing because the algorithm describes only

[Figure]

**Figure AC1.1:** Forward run with the initial conditions from the paper Sect. 4 without correction of the state and parameters. The ensemble (light green) is equally weighted and the mean is calculated from the ensemble at the last time step.

the resampling process. It requires the updated weights and the states and the time of the observation. The propagation of the ensemble and the actual weighting is not presented in this algorithm.

**Changes:** Appendix A: We corrected $u_i$ to $u_i^k$, added the application of the tuning parameter and clarified that we only show one resampling step.

**Comment:** *References: please check the reference list: many references have the doi repeated twice*

**Changes:** Thank you for noticing this. We corrected the references.

**Answers to Jasper Vrugt**

**Summary:** *This is a re-review of the paper by Berg et al. on "Covariance resampling for particle filter – state and parameter estimation for soil hydrology". I have carefully read the responses of the authors' to my earlier comments and those of the other referee. The authors' responses are professional and clearly indicate how and where changes were made and why. This certainly makes reviewing easier (as not all authors write a detailed response letter).*

**Reply:** We thank Jasper Vrugt for the second review and appreciate his additional comments and specification of his concerns, which will help to improve our manuscript. In the following we provide the answers to the comments.

**General comments**

*In my letter to the editor I wrote the following:*
*"...I can entertain/live with some of their answers to my comments, yet, I remain skeptical that the proposed method is going to work for more complex problems - and/or enlarged parameter space (current ranges are rather tight). The authors' responses to my more theoretical questions I more or less expected. They do not deny/refute important weaknesses/difficulties of their proposed resampling method. In my view the paper focuses too much on practical application without going in depth about the underlying theory. Some changes have been made to the paper to mention problems with PDF approximation, yet, I feel that the method is not properly demonstrated with the Richards' type flow model. Authors emphasize in their response letter their interest in the application - and that the PF is designed for soil hydrology (not mentioned this way) - and thus state/parameter dimensionality is not of their concern. In my view this really limits the potential impact of the paper. Right now, the paper is very much an Engineering solution - which may work well for simple problems with a model that tracks the data closely (transition density is more than adequate) and low state and parameter dimensionality. Beyond this simple case nothing else is demonstrated. What is more, the authors agree with my comment on the subjectivity of the inflation factor, gamma (see equation 21) - but their response that this variable can be estimated on the fly during assimilation is problematic as several different values of gamma will work. Some value of gamma will provide a nicer*

*convergence of the multivariate parameter and state distribution than others. But all those gamma values are equally likely; in other words, there is no metric that will help determine what value of gamma to use; so what value to use in practice? Personally, I find this troublesome - hence why I used the wording "Engineering Solution". I can live with this but this really diminishes the impact of the work. Of course, HESS is not a statistics journal, yet, research focused on methodological improvements should in my view be held to higher standards than more application oriented papers. Ideally, we would want people from other fields use our methodologies - that requires that improvements satisfy underlying statistical principles. This is not the case. I will not use this to say that the paper should not be published. I just want to express that I think that the paper could be much better if:*

*1) the authors were more concerned with "statistics", that is, satisfying/honoring important principles of convergence and approximation. The resampling method does not leave the target PDF invariant. This is something that I think we should care about as others may apply the method and present the results as if they are exactly right. This will not be the case. I see a parallel here with the poor numerics of many models used in surface hydrology. For years, this issue was denied/not investigated as people believed that the numerical error would be small compared to, say, measurement errors. Then, from 2004 papers by Kavetski, Clark, Schoups, etc. have demonstrated the effect of poor numerics on parameter and state estimates. Turned out to have a major effect; what is more, this work more or less refuted the need for global optimization methods as mass-conservative solvers turned out to produce much smoother response surfaces with a much better defined global optimum. Thus, a deeper focus on the methodology may proof very useful - and will certainly enhance tremendously the impact of the paper.*

*2) The application in the presented paper is rather weak - that is - the model (transition density) is (essentially) perfect and the parameter/state space is rather low. This is not realistic for real-world applications. This begs the question on how well the method is going to do in practice. In my view this is an important shortcoming.*

*For a hydrology paper, I can certainly accept weakness (1) above, yet then I would expect the authors to carefully examine their method; this is not the case - weakness (2) has not been addressed in the revision. I think it is important to do so - but will not object against publication if the authors do not address this. I do not want to hold up publication - as I have great respect for Prof. Roth and his co-authors. Hence my recommendation for a minor revision."*

**Reply:** Thank you for your comments.

1) It is correct that the target PDF is not invariant under resampling using the proposed method. The PDF after resampling is a superposition of the estimated target PDF and a multivariate Gaussian of the new particles. If the model and the model error were perfectly known, we agree that it would be desirable to estimate/represent the target PDF as accurately as possible. However, if the model has an unrepresented model error (not part of the forward model) or the observation error is specified incorrectly (e.g. Gaussian distribution assumed, but lognormal distributed is correct), the target PDF from Bayes' theorem is not the true PDF since the transition/observation PDF is wrong. In those cases, which are typical for more complicated situations, considering an approximation for the target PDF can be beneficial to increase the stability of the filter and to explore the state space.

For example, the ensemble Kalman filter (EnKF) is one of the most used data assimilation methods in hydrology. The EnKF assumes for all occurring PDFs Gaussian distributions and is successfully applied in hydrology (e.g. Shi et al., 2015; Man et al., 2016; Botto et al., 2018). Zhang et al. (2017) compared the particle filter with the EnKF for the application on land surface models with the result of similar performance with slight advantages of the EnKF. Through the assumption of Gaussian distributions, the EnKF can actually correct ensemble members based on the measurement values, while the particle filter only assigns a likelihood and consequently typically requires more particles than the EnKF. This shows, in the case of the EnKF, how the advantages from assuming the Gaussian distribution in the EnKF compensate the disadvantages of the approximation of the distribution.

In the presented covariance resampling method the Gaussian approximation is only applied in the resampling to generate new particles. The approximation allows to use the covariance information in the ensemble, which facilitates the generation of meaningful new particles and improves the exploration of the state space. This reduces the required number of particles in difficult situations when many particles have to be resampled. It furthermore reduces the probability of filter degeneracy on the cost that these resampled particles alter the posterior. In less difficult situations, fewer particles are resampled and the estimated posterior comes closer to the ideal distribution.

We extended the explanation of this in the manuscript (Line 5-16, Page 6).

The factor $\gamma$ is not mandatory and not required if one can afford a sufficient ensemble size. We show for our case that the covariance resampling works without applying a factor ($\gamma = 1$). However, a factor $\gamma > 1$ can strongly reduce the required number of particles. The factor helps to explore the state

and parameter space and prevents filter collapse for small effective sample sizes. We see this as an advantage of the proposed covariance resampling that it allows the easy application of the factor and offers the possibility to improve the performance. Note that the factor only applies to the resampled particles and that in less difficult situations, fewer particles are resampled and the posterior after resampling is closer to the estimated distribution. We clarify the explanation accordingly in the manuscript (see Section 6.1).

It is correct that the factor $\gamma$ is heuristic. However, it cannot be chosen arbitrarily since too large values result in overinflation that cannot be compensated by the filter anymore. We do show the results for a few values of $\gamma$ in the manuscript to give the reader an idea about the reasonable range for our example.

The factor has similarities to multiplicative inflation for the EnKF. It increases the uncertainty and prevents filter collapse for small effective sample sizes. In our example the initial state is the most challenging task for the filter. Perturbing the truth instead of the interpolation would result in an estimation without the factor and the same ensemble size. Therefore, it would be desirable to have an adaptive approach for $\gamma$ such that it is large in the beginning and reduces to 1 once the filter has a good result for the state. In cases of a model error, $\gamma$ has to increase again.

To visualize the behavior of the covariance resampling for non-Gaussian distributions, also in relation to the EnKF, we follow the comparison of deterministic and stochastic Kalman filters by Lawson and Hansen (2004).

The comparison is performed on a single analysis step. As in Lawson and Hansen (2004), the prior is a bimodal distribution. The bimodal distribution is constructed with two equally probable Gaussian distributions:

$$P(x) = \frac{1}{2\sqrt{2\pi}} \left( \exp\left[ -\frac{1}{2}(x - \overline{x})^2 \right] + \exp\left[ -\frac{1}{2}(x + \overline{x})^2 \right] \right) , \qquad (1)$$

where $\overline{x}$ is the offset of each peak from zero. In the following example, we chose $\overline{x} = 4$. For the calculation of the Kalman gain, the bimodal prior with zero mean and a variance of $\sigma_{\text{Prior}}^2 = 17$, is equal to a Gaussian with these two statistical moments ($\mathcal{N}(0, \sigma_{\text{Prior}}^2)$).

Using Eq. (1) a rather large ensemble ($N = 5000$) is generated to represent the prior. This ensemble size is used to observe the behavior of the filters without large statistical noise. The prior distribution, including the generated ensemble, is shown in Fig. AC2.1.

[Figure]

**Figure AC2.1:** Prior bimodal distribution $P(x)$ (black, Eq. (1)) and histogram of the 5000 ensemble members (green).

The analysis is calculated for an observation $d$ at the position $d = 3.5$, with three different observation errors $\sigma_{\text{Obs}}^2$. The observation errors are chosen as $\sigma_{\text{Obs}}^2 = \sigma_{\text{Prior}}^2/2$ , $\sigma_{\text{Obs}}^2 = \sigma_{\text{Prior}}^2$ and $\sigma_{\text{Obs}}^2 = 2\sigma_{\text{Prior}}^2$. Figure AC2.2 shows the resulting analytical posterior distributions calculated using Bayes' theorem and the posterior ensemble of the EnKF, the particle filter with covariance resampling (PFCR) and the PFCR using $\gamma = 1.2$.

In the case of an accurate observation $\sigma_{\text{Obs}}^2 = \sigma_{\text{Prior}}^2/2$ (left), the observation determines the right mode of the bimodal prior as the posterior. The EnKF corrects the states towards the observation and the posterior becomes approximately Gaussian but with a mean shifted to lower values and a larger variance compared to the truth. The PFCR can describe the posterior in this particular case excellently. It does not need to shift the ensemble members of the left mode to the observation like the EnKF, instead almost 50 % of the ensemble members are dropped and resampled. This resampling is effective because the covariance resampling samples from a Gaussian distribution and the posterior is approximately Gaussian.

In the case of a less accurate observation $\sigma_{\text{Obs}}^2 = \sigma_{\text{Prior}}^2$ (middle), the observation information is less dominant and the bimodal structure of the prior is partly visible. For the EnKF, the ensemble members of the left mode are shifted towards the observations such that the posterior becomes unimodal. The bimodal structure cannot be described by the EnKF properly. The PFCR can sample both posterior peaks with the retained ensemble. The new particles, however, are generated with a mean that lies in between both peaks, such that some of the new particles are located between the modes.

[Figure]

**Figure AC2.2:** Analysis ensemble (green) for EnKF (second row), particle filter with covariance resampling (PFCR, third row) and PFCR with $\gamma = 1.2$ (bottom row) for a bimodal prior (Fig. AC2.1), for observations (top row, red) with different observation errors: $\sigma^2_{\text{Obs}} = \sigma^2_{\text{Prior}}/2$ (left column), $\sigma^2_{\text{Obs}} = \sigma^2_{\text{Prior}}$ (middle column) and $\sigma^2_{\text{Obs}} = 2\sigma^2_{\text{Prior}}$ (right column). The true posterior (black) is calculated with Bayes' theorem. In case of the PFCR, 49.7 % (left), 32.6 % (middle) and 10.0 % (right) particles are resampled.

The deviations from the truth show how the PFCR alters the posterior distribution in this case.

For $\sigma^2_{\mathrm{Obs}} = 2\sigma^2_{\mathrm{Prior}}$ (right), the posterior is similar to the prior with a less likely left mode. The large uncertainty makes it impossible to confidently decide in which peak the truth lies. The EnKF retains the bimodal distribution of the prior because the Kalman gain becomes small if the observation error is large, which leads to a small correction in the analysis step. The particles have approximately equal weights because of the large observation error. Therefore, only $10.0\,\%$ of the particles are resampled which reduces the effect that particles are generated in between the peaks as in the case $\sigma^2_{\mathrm{Obs}} = \sigma^2_{\mathrm{Prior}}$ (middle).

The factor $\gamma$ has only a small effect in this example. Compared to the EnKF the PFCR describes the real PDF more accurately with only minor deviations from the true PDF.

2) The joint estimation of states and parameters in soil hydrology based on the Richards equation remains a challenging task. Applications with particle filters for state and parameter estimation directly to the Richards equation remain few (e.g. Montzka et al., 2011; Manoli et al., 2015), and estimate 1-5 parameters. We focus on a one-dimensional synthetic case to introduce the covariance resampling and show its behavior in detail.

We agree that the behavior in presence of model errors is relevant. We included the exemplary study of one error, the boundary condition, in the previous revision to show that the PFCR is not limited to perfectly known transition densities. However, an comprehensive investigation of model errors in general is far beyond the scope of this paper.

We improved and clarified the description of the presented example (see also reply to comment 6).

**Changes:** Section 6.1: We added more explanation of the factor $\gamma$, its influence on the estimated pdf and its relation to the multiplicative inflation. Line 5-16, Page 6: We extended the explanation of the influence of the covariance resampling on the estimated pdf.

**Specific comments**

**Comment:** *1. Table 1 - alpha values should have units of 1/m? Or are authors presenting values of 1/alpha instead? That explains the very large*

**Table AC2.1:** True Mualem-van Genuchten parameters and range of the uniformly distributed initial guess.

| Parameter | Truth | Lower | Upper | Lower$_{\text{prev.}}$ | Upper$_{\text{prev.}}$ |
|---|---|---|---|---|---|
| $n_1$ $[-]$ | 2.28 | 1.5 | 6.0 | 2.2 | 3.5 |
| $n_2$ $[-]$ | 1.89 | 1.5 | 6.0 | 1.8 | 3.2 |
| $\alpha_1$ $[\text{m}^{-1}]$ | -12.4 | -14 | -5 | -14 | -12 |
| $\alpha_2$ $[\text{m}^{-1}]$ | -7.5 | -10.5 | -2.5 | -10.5 | -6.5 |
| $\log_{10}(K_{w,1})$, $K_w$ in $[\text{m s}^{-1}]$ | -4.40 | -10 | -3 | -7 | -4 |
| $\log_{10}(K_{w,2})$, $K_w$ in $[\text{m s}^{-1}]$ | -4.91 | -10 | -3 | -7.5 | -4 |

*values of -12.4 and -7.5 for $\alpha_1$ and $\alpha_2$ as the air-entry value?*
**Reply:** Thank you for noticing this. The unit is m$^{-1}$.
**Changes:** Table 1, Page 10: We corrected the unit.

**Comment:** *2. Parameter ranges are rather narrow. What would happen if we enlarge n to say 1.1 - 6?*
**Reply:** We increased the parameter range for $K_w$, $n$ and $\alpha$ for both layers (see Table AC2.1) to further investigate the behavior. Note, that for $n$ we only increased the range to 1.5-6.0. The simplified Mualem-van Genuchten parametrization used here is only well-behaved for $n > 2$ (Ippisch et al., 2006) and undefined for $n \leq 1$. For small $n$ the performance of the numerical solver decreases significantly such that $n > 1.1$ is necessary. Since the covariance resampling generates new ensemble members from a Gaussian distribution, it can occur that $n < 1.1$. In this case we set the value to $n = 1.1$. We used the interval $1.5 - 6.0$ to reduce the occurrence of this case.

It was necessary to use more ensemble members for a converging result. In this case we used 300 ensemble members. The final state is shown in Fig. AC2.4 and the conductivity function Fig. AC2.3. Both are in good agreement with the synthetic truth.

**Changes:** Line 1-3, Page 11: We added the sentences: "The filter can also estimate the state and parameters for an extended range." and " Increasing the initial uncertainty of the parameters, increases the complexity of the problem and the filter needs more ensemble member to converge." to the manuscript.

**Comment:** *3. Table 1: Are the log(Kw) values natural log-values? I do not think so - otherwise we get values of exp(-4) \* 60 \* 60 \* 24 ≈ 1600 m/day. So they must be log10 right? With log10(Kw) values between -7 and -4 the range of Kw is between 0.0086 and 8.64 m/day. The lower bound of 0.86*

[Figure]

**Figure AC2.3:** Conductivity function $K(h_m)$ for (a): layer 1 and (b): layer 2. In this function all estimated parameters are represented. The initial 95 %-quantile of the ensemble (light green) with the mean (green) are shown. The truth (dashed black) is almost congruent with the estimated mean (orange), such that only the 95 %-quantile of the final ensemble (light orange) is visible.

[Figure]

**Figure AC2.4:** Final water content state after the assimilation run. The truth (dashed black) is almost congruent with the estimated mean (orange), such that only the 95 %-quantile of the ensemble (light orange) is visible. Notice that even the maximal difference is well below current measurement capabilities.

*cm/day may still be considered large for say fine textured soils. Therefore I would suggest using a range of log10(Kw) between -10 and -4 or so. Point is - the parameter space is narrow - this simplifies tremendously inference - but may not necessarily demonstrate convergence properly; how does the filter work with enlarged parameter spaces that encompass a larger ranges of soils?*
**Reply:** Thank you for pointing this out. $log_{10}$ is correct. For the application with a larger parameter spread please refer to comment 2.
**Changes:** Table 1, Page 10: We changed *log* to $log_{10}$ and refer to it as decadic logarithm to clarify this (Line 8, Page 10).

**Comment:** *4. Line 18: generic algorithm? Or genetic algorithm? Generic algorithm would not make sense as this has been presented before in other papers - for example Particle-DREAM. A genetic algorithm on the contrary will not maintain detailed balance unless they use the Differential Evolution variant with Metropolis step as done in DE-MC and DREAM. I have to read this new paper to understand/see what has been done and how this differs from previous DE-MC/DREAM MCMC work.*
**Reply:** Thank you for noticing this. It is a genetic algorithm.
**Changes:** Line 17, Page 2: Changed "generic" to "genetic" algorithm.

**Comment:** *5. Figure 11: Those 100 ensemble members - are they randomly*

*chosen? Or are these the "best" members; I would be interested to see all the members as "visibility" is not an argument here. You cannot see the 100 different lines anyway (much fewer in practice as particles have many copies). Instead of using Lines you can color the 95 % ranges of all particles.*

**Reply:** The ensemble members in Figure 11 and Figure 12 had been randomly chosen. We follow your idea and show the 95 %-quantile of the ensemble.

**Changes:** Throughout the manuscript: Changed all figures with an ensemble to show the 95 %-quantile.

**Comment:** *6. In section 6.3, the authors attempt to address the model error by perturbing the forcing ( = boundary conditions). This is not really a model error but measurement error. Indeed, the PF should rapidly address ( = correct for) those errors in a few assimilation steps as this entails conservation of mass. What is much more difficult is to address actual model errors due to an incorrect/absent process representation. Those errors lead to much more complicated and predictable residual patterns, making state and certainly parameter estimation more difficult. Will produce parameter estimates that may compensate at least somewhat for the model errors. I think it is important to address, at least in writing, this difference between forcing data errors and model errors. These two errors are not equal.*

**Reply:** We agree that there are different types of model errors. In a real case, an error in the forcing would be a measurement error. Using it in a synthetic study, the changed boundary condition is similar to an additional source/sink term. This leads to a wrong transition density (compared to the truth) since the observations are still generated using the old boundary condition. In the presented case, we have a continuous bias towards lower water content which the filter continuously needs to address. As you mentioned, this leads to a bias in the parameter estimates to compensate the error in the boundary condition. The logarithmic scale in the conductivity plot reduces the visibility of this bias.

**Changes:** Line 29-31, Page 18: We clarified the effect of the changed boundary condition.
Line 11-12, Page 19: We point out the bias in the parameter.

[revised manuscript text omitted]